# A Lin28 homologue reprograms differentiated cells to stem cells in the moss *Physcomitrella patens*

Chen Li[1,2,†], Yusuke Sako[1,3], Akihiro Imai[1,3,†], Tomoaki Nishiyama[3,4], Kari Thompson[1,3,5], Minoru Kubo[1,3,†], Yuji Hiwatashi[1,2,†], Yukiko Kabeya[1], Dale Karlson[5], Shu-Hsing Wu[6], Masaki Ishikawa[1,2], Takashi Murata[1,2], Philip N. Benfey[7], Yoshikatsu Sato[1,3,†], Yosuke Tamada[1,2] & Mitsuyasu Hasebe[1,2,3]

Both land plants and metazoa have the capacity to reprogram differentiated cells to stem cells. Here we show that the moss *Physcomitrella patens* Cold-Shock Domain Protein 1 (PpCSP1) regulates reprogramming of differentiated leaf cells to chloronema apical stem cells and shares conserved domains with the induced pluripotent stem cell factor Lin28 in mammals. PpCSP1 accumulates in the reprogramming cells and is maintained throughout the reprogramming process and in the resultant stem cells. Expression of *PpCSP1* is negatively regulated by its 3′-untranslated region (3′-UTR). Removal of the 3′-UTR stabilizes *PpCSP1* transcripts, results in accumulation of PpCSP1 protein and enhances reprogramming. A quadruple deletion mutant of *PpCSP1* and three closely related *PpCSP* genes exhibits attenuated reprogramming indicating that the *PpCSP* genes function redundantly in cellular reprogramming. Taken together, these data demonstrate a positive role of PpCSP1 in reprogramming, which is similar to the function of mammalian Lin28.

[1] National Institute for Basic Biology, Division of Evolutionary Biology, Okazaki 444-8585, Japan. [2] Department of Basic Biology, School of Life Science, SOKENDAI (The Graduate University for Advanced Studies), Okazaki 444-8585, Japan. [3] ERATO, Hasebe Reprogramming Evolution Project, Japan Science and Technology Agency, Okazaki 444-8585, Japan. [4] Advanced Science Research Center, Institute for Gene Research, Kanazawa University, Kanazawa 920-0934, Japan. [5] Division of Plant and Soil Sciences, West Virginia University, Morgantown, West Virginia 26506, USA. [6] Institute of Plant and Microbial Biology, Academia Sinica, Taipei 11529, Taiwan. [7] Department of Biology and Howard Hughes Medical Institute, Duke University, Durham, North Carolina 27708, USA. † Present addresses: School of Pharmacy, Hubei University of Medicine, Shiyan city, Hubei 442000, China (C.L.); Faculty of Life Sciences, Hiroshima Institute of Technology, Hiroshima 731-5193, Japan (A.I.); Institute for Research Initiatives, Nara Institute of Science and Technology, Nara 630-1092, Japan (M.K.); School of Food, Agricultural and Environmental Sciences, Miyagi University, Sendai 982-0215, Japan (Y.H.); Institute of Transformative Bio-Molecules (WPI-ITbM), Nagoya University, Nagoya 464-8602, Japan (Y.S.). Correspondence and requests for materials should be addressed to Y.T. (email: tamada@nibb.ac.jp) or to M.H. (email: mhasebe@nibb.ac.jp).

Stem cells can self-renew and produce cells to be differentiated during development[1–4]. On the other hand, differentiated cells can change their cell fate to stem cells under certain conditions in both land plants and metazoa[3,4]. In flowering plants, differentiated cells can form undifferentiated cell masses called callus. With the addition of the appropriate phytohormones they can regenerate shoot and root meristems including stem cells, as was first shown with carrot in 1958 (ref. 5). Several genes have been shown to be involved in the formation of callus or regeneration of stem cells in *Arabidopsis thaliana* (*Arabidopsis*). Overexpression of a plant-specific AP2/ERF transcription factor ENHANCER OF SHOOT REGENERATION 1 (ESR1)/DORNROESCHEN (DRN) promotes the formation of shoot meristems from callus[6]. Induction of another AP2/ERF transcription factor WOUND INDUCED DEDIFFERENTIATION 1 (WIND1) enhances callus formation without exogenous hormones[7]. In bryophytes, differentiated cells have a remarkable ability of being reprogrammed into stem cells without callus formation. In the moss *Physcomitrella patens* (*Physcomitrella*), wounding can induce the transition from differentiated leaf cells into proliferating chloronema stem cells without any exogenous phytohormones[8,9]. To understand the molecular mechanisms underlying this reprogramming, transcriptome analysis was performed during the reprogramming[10] and several factors were identified as playing a role in the process. For instance, Cyclin-Dependent Kinase A (CDKA) activation is essential for cell cycle re-entry during reprogramming[9]. *WUSCHEL-related homeobox 13-like* (*WOX13L*) genes are required for the initiation of tip growth during stem cell formation[11].

In mammals, the induction of four factors is sufficient to reprogram somatic cells to pluripotent stem cells. Oct4, Sox2, cMyc and Klf4 were first reported as induced pluripotent stem cell (iPSC) factors able to reprogram mouse fibroblast cells into pluripotent stem cells[12]. Later, the same factors were applied to human fibroblast cells to generate iPSCs[13]. At the same time, another set of pluripotency factors Oct4, Sox2, Nanog and Lin28 was identified, which could successfully induce pluripotent stem cells from human fibroblast cells[14]. So far, factors belonging to the same gene family and functioning in reprogramming from differentiated cells to stem cells have not been identified between land plants and metazoa. Therefore, it is still unknown whether plants and animals use similar mechanisms for the reprogramming from differentiated cells to stem cells.

Here, we report that *P. patens* Cold-Shock Domain Protein 1 (PpCSP1), which shares highest sequence similarity and domain structure with Lin28 in metazoa, enhances reprogramming in *Physcomitrella*. PpCSP1 accumulates in the reprogramming cells. *PpCSP1* expression is negatively regulated by its 3′-untranslated region (3′-UTR). When the 3′-UTR is removed, *PpCSP1* transcripts increase and the reprogramming is enhanced. Deletion of *PpCSP1* and three closely related *PpCSP* genes causes attenuated reprogramming, demonstrating a positive and redundant function of PpCSPs in the reprogramming.

## Results

**PpCSP1 shares conserved domains with Lin28.** Cold-shock domain proteins (CSPs) were first identified in bacteria as proteins expressed under cold-shock conditions[15], and were later implicated in the process of cold acclimation in flowering plants as CSP transcripts accumulate after cold treatment in *Arabidopsis* and wheat[16–18]. The cold-shock domain (CSD) is highly conserved in bacteria, land plants and metazoa. CSD possesses nucleic acid binding activity and is capable of binding to single-stranded DNA/RNA and double-stranded DNA[19]. To better understand the evolution of CSPs, we investigated the function of the *PpCSP1* gene in *Physcomitrella* since no previous study had focused on CSPs in non-flowering plants[20]. To characterize the expression pattern of PpCSP1, we generated a PpCSP1-Citrine fusion protein line (nPpCSP1-Citrine-nosT; Supplementary Fig. 1a,b). Using live imaging, we detected predominant PpCSP1-Citrine signals in chloronema and caulonema apical stem cells, which self-renew and produce cells that differentiate into chloronema and caulonema cells, respectively (Fig. 1a,b and Supplementary Fig. 1c). The signals were also detected in chloronema and caulonema side branch initial cells, which are typically destined to become chloronema apical stem cells (Fig. 1a,b). These results suggested the possible involvement of PpCSP1 in stem cell maintenance and in the reprogramming of differentiated chloronema and caulonema cells to chloronema apical stem cells[8]. In addition to CSD, a search for conserved domains[21] (www.ncbi.nlm.nih.gov/Structure/cdd/wrpsb.cgi) in PpCSP1 identified a provisional domain PTZ00368 (universal minicircle sequence-binding protein), which is comprised of two CCHC zinc-finger domains (Fig. 1c). Most plant CSPs and some animal CSPs also have CCHC zinc-finger domains but bacteria CSPs do not. We then performed BLASTP searches using the PpCSP1 sequence as a query to identify proteins related to PpCSP1. Lin28 proteins were the top hits when the BLAST searches were performed against the database of metazoa, including *Homo sapiens*, *Mus musculus* and *Caenorhabditis elegans*. The Lin28 proteins, one of which is an iPSC factor in human, share one CSD and two CCHC zinc-finger domains with PpCSP1 (ref. 22 and Fig. 1c). We subsequently inferred phylogenetic relationships of PpCSP1 and other proteins with these three domains using the maximum likelihood tree reconstruction method of RAxML[23]. Although the low resolution of the phylogenetic tree did not enable us to examine whether PpCSP1 is orthologous or paralogous to Lin28 (Fig. 1d), PpCSP1 and Lin28 should be homologous because of the shared domains and these results led us to investigate whether PpCSP1 plays a role similar to Lin28 in reprogramming differentiated cells to stem cells.

**PpCSP1 mRNA and protein accumulate during reprogramming.** To investigate the function of *PpCSP1* in reprogramming, we cut gametophore leaves and cultivated them on a medium without phytohormones[9]. Gametophores are shoots formed in the haploid generation (Supplementary Fig. 2a). When a differentiated leaf is excised from a gametophore, leaf cells facing the cut change to chloronema apical stem cells with tip growth and divide ∼30 h after excision[9] (Supplementary Fig. 2b). A chloronema apical stem cell divides to regenerate itself and form a chloronema subapical cell. Therefore, chloronema apical stem cells fulfil the definition of a stem cell: they self-renew and give rise to cells that go on to differentiate. All leaf cells with tip growth behave as chloronema apical stem cells[9] and this acquisition of a new fate is the most reliable sign of the reprogramming at present. To examine the spatiotemporal expression pattern of the PpCSP1 protein in cut leaves, we removed the DNA fragment containing the nopaline synthase polyadenylation signal (nosT)[9] and the neomycin phosphotransferase II (nptII)[24] expression cassette from the nPpCSP1-Citrine-nosT line by transiently expressing Cre recombinase[25]. As a result, the native 3′-UTR was fused to the *PpCSP1-Citrine* coding sequence (CDS) (nPpCSP1-Citrine-3′-UTR line; Supplementary Fig. 1a). During the reprogramming process, Citrine signals specifically increased in leaf cells facing the cut just after excision (Fig. 2a,b; Supplementary Movie 1; and Supplementary Fig. 1d). The Citrine signals increased continuously until tip growth started. Even though the Citrine signal increased in most edge cells, fewer than half of the edge cells protruded. These observations suggest that other factors

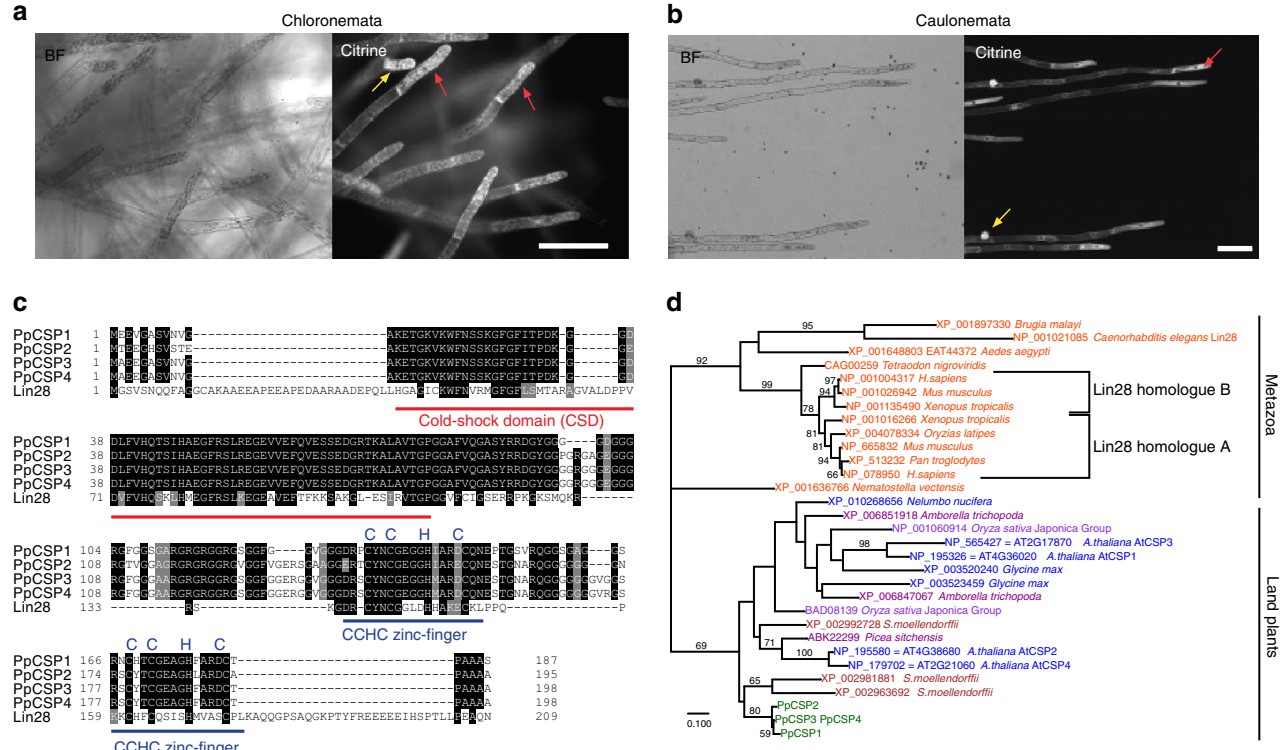

**Figure 1 | PpCSP1 that shares conserved domains with Lin28 is expressed in protonema apical stem cells.** (**a,b**) Bright-field (BF) and fluorescence (Citrine) images of chloronemata (**a**) and caulonemata (**b**) of the nPpCSP1-Citrine-nosT #136 line. Red and yellow arrows indicate apical stem cells and side branch initial cells, respectively. (**c**) Alignment of the amino acid sequences of PpCSPs and human Lin28 proteins. PpCSPs and human Lin28 proteins were predicted to contain one CSD (red line) and two CCHC zinc-finger domains (blue lines). Black and grey shades indicate identical amino acids and amino acids with similar characters to the consensus amino acid, respectively. (**d**) Phylogeny of PpCSP1, Lin28 and related proteins, with a cold-shock domain and zinc-finger domains. The maximum likelihood tree was constructed using amino acid sequences of the proteins. The wag model of amino acid substitution was used. Branch lengths are proportional to the number of substituted residues. Bootstrap probability >50% is indicated on the branches (estimated by 1,000 resampling). The accession numbers and species names are indicated. Colour of the OTU represents the phylogenetic position: Orange, metazoans; blue, eudicots; light purple, monocots; dark purple, other seed plants including gymnosperms and basal angiosperms; green, bryophytes; brown, lycophytes. This is an unrooted tree. The left-most node was chosen for the best match of organism phylogeny. Mammalian *Lin28* genes used for the iPSC reprogramming are included in the 'Lin28 homologue A'. Scale bars, 100 μm (**a,b**). The scale bar represents the number of amino acid substitutions per site in **d**.

unevenly distributed in the edge cells are also involved in reprogramming. After the protrusion, PpCSP1-Citrine signals localized more conspicuously at the phragmoplast than other parts in the cytosol. The signals were dispersed in the cytosol after cytokinesis with remaining signals at the cell septum. The signals at the phragmoplast decreased during subsequent cell divisions of chloronema apical stem cells (Supplementary Fig. 3 and Supplementary Movie 2). These indicate that PpCSP1 protein predominantly accumulates in the leaf cells facing the cut, accumulates during reprogramming, gradually decreases after reprogramming, and is maintained in stem cells. In the growing protonemata, PpCSP1-Citrine was continuously expressed in apical stem cells during the entire cell cycle (Supplementary Movie 3). When side branch cells initiated, PpCSP1-Citrine signals increased during protrusion and localized at the phragmoplast. The signals at the phragmoplast decreased during subsequent cell divisions (Supplementary Movie 3). In addition, PpCSP1 was expressed in proliferating cells in gametophore apices where both stem cells and proliferating non-stem cells exist[8] (Supplementary Fig. 1e). This is reminiscent of Lin28, which regulates cell cycles in stem cells[26,27]. PpCSP1-Citrine localized in the cytosol but not in the nucleus (Fig. 2c). Because of the presence of the CSD and zinc-finger domains, it is plausible that PpCSP1 functions as an

RNA-binding protein to regulate mRNA maturation, stability, or translation in the cytosol in a manner similar to that reported for other CSPs[18,19], including Lin28 and related proteins in metazoa.

To analyse the promoter activity of *PpCSP1*, we made a transcriptional fusion (PpCSP1pro:LUC), in which the coding sequence of *luciferase* (*LUC*)[28] is driven by the 1.8 kb *PpCSP1* promoter. This construct was integrated into the PIG1 neutral site[29,30] of the nPpCSP1-Citrine-3′-UTR background line (Supplementary Fig. 2c,d). With this dual reporter construct (PpCSP1pro:LUC nPpCSP1-Citrine-3′-UTR), we are able to simultaneously monitor promoter activity and protein accumulation at a single-cell level (Fig. 2d–f and Supplementary Fig. 2e–j; and Supplementary Movie 4). Time-lapse imaging showed LUC signals from PpCSP1 promoter activity increasing after excision (Fig. 2e). In edge cells that would later protrude, the intensities maximized at ~12 h and were maintained with some fluctuation (Fig. 2e, left). However, the rates of increase and the maxima of the intensities varied among cells. In edge cells that never protruded, LUC signals initially increased but were not maintained as they were in the protruded edge cells (Fig. 2e, right). PpCSP1-Citrine levels in edge cells that would protrude continued to increase from 24 to 36 h, until these cells divided (Fig. 2f). In edge cells that never protruded, Citrine accumulation

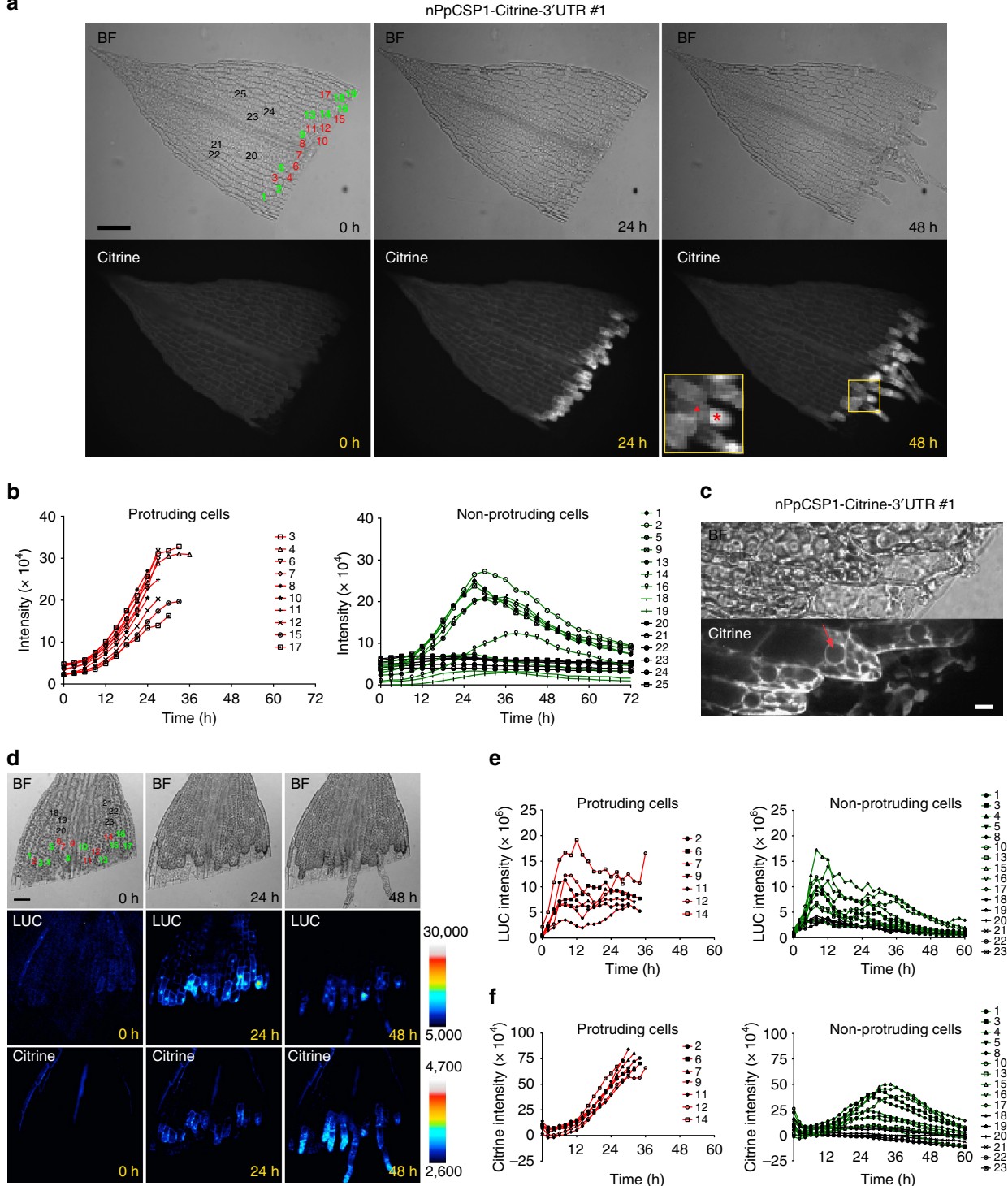

**Figure 2 | PpCSP1 is induced in the process of reprogramming.** (**a**) Expression pattern of PpCSP1-Citrine in an excised leaf of nPpCSP1-Citrine-3′-UTR #1 line. Bright-field (BF) and fluorescent (Citrine) images at 0, 24 and 48 h after cutting are shown. Inset red star and triangle indicate a distal chloronema apical stem cell and a proximal chloronema cell, respectively. All edge cells and several non-edge cells were numbered for quantitative analysis in **b**. See also Supplementary Movie 1. (**b**) The intensity of the Citrine signals in each cell of an excised leaf of nPpCSP1-Citrine-3′-UTR #1 (1–25 correspond to cells in the top panel of **a**). Red and green lines indicate the signal intensity in edge cells that were and were not reprogrammed into stem cells, respectively. Black lines indicate the signal intensity in non-edge cells that were not reprogrammed into stem cells. (**c**) PpCSP1-Citrine fusion protein localization in excised leaf cells 24 h after cutting of the nPpCSP1-Citrine-3′-UTR #1 line. Red arrow indicates the nucleus. (**d**) *PpCSP1* promoter activity and the protein accumulation during the reprogramming. Bright-field (top), luciferase (middle) and Citrine images (bottom) of an excised leaf of the PpCSP1pro:LUC nPpCSP1-Citrine-3′-UTR #2 line at 0, 24 and 48 h after cutting. Calibration bars were shown for pseudo-colour images of LUC and Citrine, respectively. All edge cells and several non-edge cells are numbered for **e**,**f**. See also Supplementary Movie 4. (**e**,**f**) The intensity of luciferase (**e**) and Citrine (**f**) signals in each cell (indicated by 1–23 in the top left of **d**) in an excised leaf. Red and green lines indicate the signal intensity in edge cells that were and were not reprogrammed into stem cells, respectively. Black lines indicate the signal intensity in non-edge cells that were not reprogrammed into stem cells. Scale bars, 100 μm (**a**); 10 μm (**c**); and 50 μm (**d**).

reached a maximum in 24–36 h and then gradually declined, which is consistent with the changes in promoter activity (Fig. 2e). The smaller variation in protein levels than in promoter activity in cells that eventually protrude (Fig. 2e,f, left) suggests the potential involvement of post-transcriptional regulation or a difference in stability of the transcripts and proteins of *PpCSP1*.

***PpCSP1* is negatively regulated through its 3′-UTR.** *Lin28* is negatively regulated by microRNA (miRNA) *let-7* (refs 31–34), which directly binds to *Lin28* transcripts at the 3′-UTR leading to the degradation of *Lin28* transcripts[31]. In the *Physcomitrella* genome, we could not identify a miRNA similar to *let-7* (refs 35–38). However, the 3′-UTR of *PpCSP1* is 623 bp, which is longer than the median length (334 bp) of 3′-UTRs in the *Physcomitrella* v1.6 genome sequence[39]. This suggests that regulatory elements could be located in the 3′-UTR. To determine if the 3′-UTR of *PpCSP1* is involved in regulating transcript abundance, we performed 5′-digital gene expression (5′-DGE) analysis[10] in the nPpCSP1-Citrine-3′-UTR and nPpCSP1-Citrine-nosT lines, in which the 3′-UTR is separated from the *PpCSP1*-coding region by the nosT and the nptII expression cassette (Supplementary Fig. 1a). We compared these results to previously published 5′-DGE data of leaf cut experiments[10] (Fig. 3a). In the 5′-DGE analysis, ∼25-bp cDNA fragments at the 5′-ends of polyadenylated RNAs are sequenced. The tags in the 5′-UTR or CDS represent RNA molecules that are not cut in the 3′-UTR, while tags in the 3′-UTR represent RNAs that are cut or undergoing degradation. The number of tags in the *PpCSP1* 5′-UTR or CDS tended to increase after the leaf cut and nPpCSP1-Citrine-nosT had a generally higher value than nPpCSP1-Citrine-3′-UTR (6.6-fold in median). In wild-type and nPpCSP1-Citrine-3′-UTR lines, more sequenced tags were mapped to the 3′-UTR than the 5′-UTR or CDS, while in the nPpCSP1-Citrine-nosT line more tags were mapped to the 5′-UTR or CDS than to the exogenous 3′-UTR of nosT (Fig. 3a). These data suggest that the 3′-UTR of *PpCSP1* is a degradation target similar to that of *Lin28*, or has a weak polyadenylation signal.

To examine the activity of the 3′-UTR, independent of its original genomic context, we generated constructs with a constitutively active elongation factor-1α (*EF1α*) promoter[40] -driven *sGFP*[41], fused to either the *PpCSP1* 3′-UTR or nosT, introduced into the PTA1 neutral site[40] (Supplementary Fig. 4a–c). sGFP intensity in the EF1αpro:sGFP-nosT line increased in all of the examined leaf cells during reprogramming after cutting (Fig. 3b,c and Supplementary Fig. 4e–h). The increase of activity was more conspicuous in edge cells than in non-edge cells (Fig. 3c). On the other hand, in the 3′-UTR-fused line, cellular signals of both edge and non-edge cells (Fig. 3d,e and Supplementary Fig. 4i–l) were ∼10 times weaker than those in the nosT-fused line (Fig. 3c,e and Supplementary Fig. 4e–l). To examine the degradation activity of the 3′-UTR under unwounded conditions, sGFP signals were compared in protonemata and gametophores between the two lines (Fig. 3f,g). In gametophores and protonemata, signals of the 3′-UTR-fused line were weaker than those in the nosT-fused line as in the reprogramming process. Reverse transcriptase-quantitative PCR (RT-qPCR) determined that transcript levels of *sGFP* were $67.3 \pm 1.5$-fold (mean ± s.d., $n = 3$) and $57.2 \pm 3.1$-fold (mean ± s.d., $n = 3$) higher in the nosT than the 3′-UTR line in gametophores and protonemata, respectively. These results indicate that the *PpCSP1* 3′-UTR contains negative regulatory signals that function, independently of the *PpCSP1* promoter, during the reprogramming process in cut leaves, as well as during regular development.

**PpCSP1 does not appear to be regulated by a microRNA.** miRNAs-evolved independently in land plants and metazoa[42–44]. However, some similarities exist between these two linages, such as conserved components like Dicer/Dicer-like and Argonaute proteins[42]. In addition, two possible *Arabidopsis* miRNAs (miRNA854 and miRNA855) were identified to be shared between land plants and metazoa and had binding sites within the 3′-UTR of the target mRNA[45]. To test whether a similar miRNA-associated regulation as *let-7* miRNA regulates *Lin28*, we made a deletion series of the *PpCSP1* 3′-UTR fusing each fragment after the stop codon of the *sGFP* reporter gene driven by the constitutive rice *Actin 1* promoter[46,47] (Fig. 3h). These constructs were transiently introduced into gametophore leaf cells by particle bombardment and co-bombarded with a fragment containing the *monomeric Red Fluorescent Protein 1* (*mRFP*) gene[48] driven by the same *Actin 1* promoter for normalization (Fig. 3h,i). The linear correlation of the sGFP and mRFP signals in the transformed cells was confirmed (Supplementary Fig. 4d). In comparison to the control (no UTR), signal intensities of sGFP fused with 623-, 500-, 400-, 300- and 200-bp 3′-UTR fragments decreased to 11.8, 15.7, 29.4, 54.9 and 74.5%, respectively (Fig. 3j). This gradual reduction suggests that several different regions in the 3′-UTR serve as targets for the negative regulation. We subsequently searched candidate miRNAs using the 3′-UTR as a query in the psRNATarget website (http://plantgrn.noble. org/psRNATarget/)[49] and analysed small RNAs at the *PpCSP1* locus in Plant Small RNA Genes WebServer (https:// plantsmallrnagenes.psu.edu/cgi-bin/Ppatens_Locus_Reporter)[35]. However, we could not find any miRNA-targeting sequences in the 3′-UTR. In the future, additional studies such as genome-wide mRNA-protein interaction analysis[50], will be needed to fully understand the molecular mechanisms of the degradation function of the *PpCSP1* 3′-UTR.

**Increase of *PpCSP1* transcript levels enhances reprogramming.** Having determined that the 3′-UTR has a degradation function, we quantified transcript levels in the nPpCSP1-Citrine-nosT line, and compared them with the nPpCSP1-Citrine-3′-UTR line and wild type. Using RT-qPCR we found that transcript levels were $6.0 \pm 2.9$-fold (mean ± s.d., $n = 3$) and $9.9 \pm 2.5$-fold (mean ± s.d., $n = 3$) higher in the nPpCSP1-Citrine-nosT line as compared with the nPpCSP1-Citrine-3′-UTR line and wild type at 0 h after leaf cutting, respectively. These results are in agreement with the 5′-DGE analysis as the tag counts in nPpCSP1-Citrine-3′-UTR were not drastically different when compared with wild type. Collectively, these results indicate that transcript levels of *PpCSP1* increased in the nPpCSP1-Citrine-nosT line.

As the *PpCSP1* transcript level is ∼10-fold higher in the nPpCSP1-Citrine-nosT line, we found that this transcript increase results in protruding non-edge cells (Fig. 4a–c). However, only edge cells protrude in wild type (Supplementary Fig. 2b). We calculated percentages of excised leaves with at least one protruding non-edge cell in wild-type, nPpCSP1-Citrine-nosT, and nPpCSP1-Citrine-3′-UTR lines (Fig. 4a,b). While the percentages of excised leaves with protruding edge cells did not differ among these lines (Fig. 4a), those with protruding non-edge cells significantly increased in nPpCSP1-Citrine-nosT (Fig. 4b). Moreover, some non-edge cells of nPpCSP1-Citrine-nosT exhibited stronger Citrine signals than surrounding cells, some of which were reprogrammed to stem cells (Fig. 4c,d; Supplementary Fig. 5a–d; and Supplementary Movie 5), while Citrine signals of nPpCSP1-Citrine-3′-UTR lines were detected in cells at the cut edge but not in non-edge cells (Fig. 2a).

To confirm the increase in protruding non-edge cells in the nPpCSP1-Citrine-nosT line, we produced a PpCSP1pro:

PpCSP1-Citrine line. In this construct, the *PpCSP1* promoter, *PpCSP1* CDS and *Citrine* gene were inserted into the neutral PTA1 site (Supplementary Fig. 5e,f), which enabled us to visualize increased PpCSP1-Citrine levels. RT-qPCR analysis indicated that

transcript levels of *PpCSP1* were $15.5 \pm 3.7$-fold (mean $\pm$ s.d., $n = 3$) higher in PpCSP1pro:PpCSP1-Citrine as compared with wild type at 0 h after leaf cutting. Spatiotemporal patterns of Citrine signals and protruding cells in the PpCSP1pro:PpCSP1-

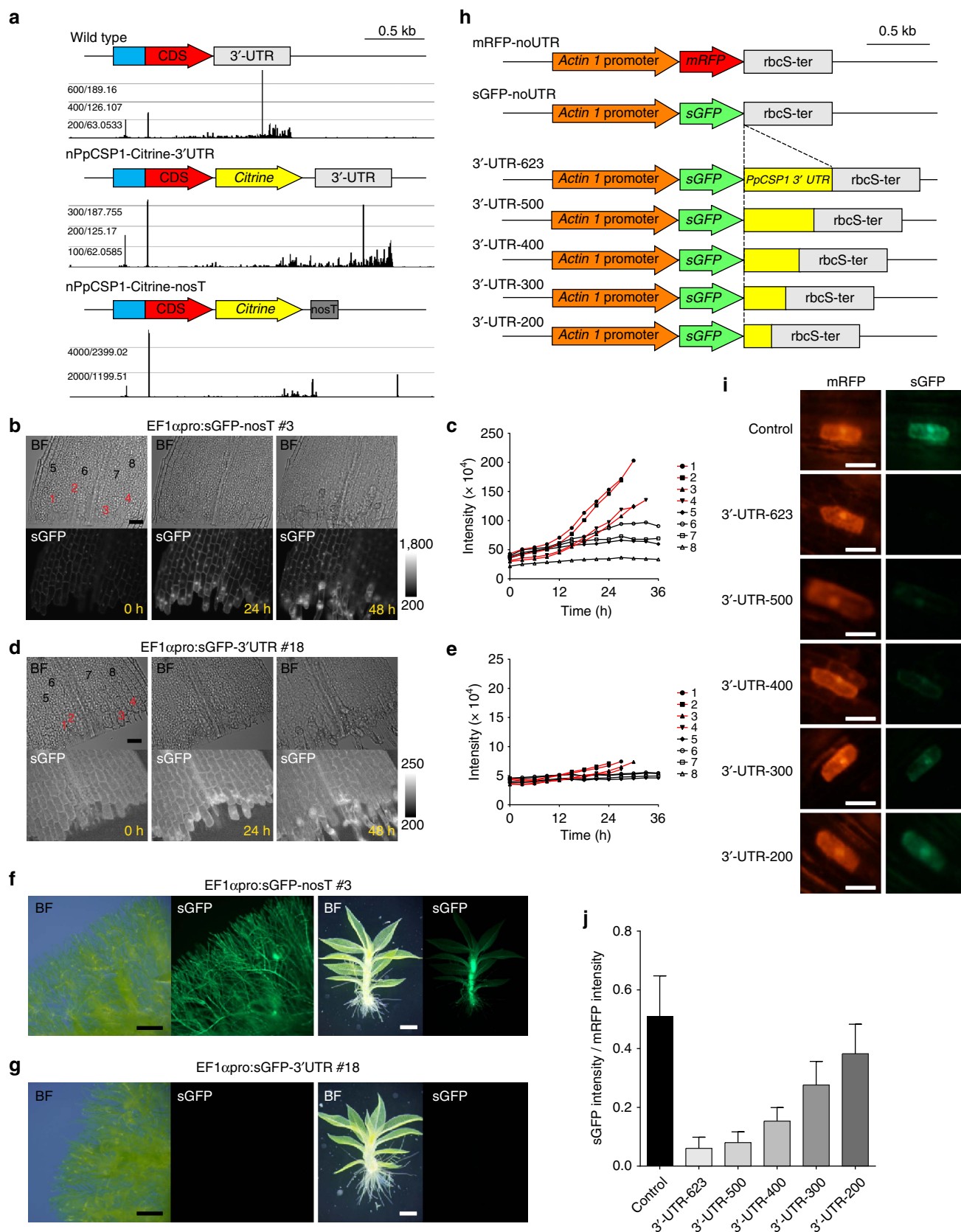

Citrine line were similar to those of the nPpCSP1-Citrine-nosT line (Fig. 4e–g). We conclude that the protruding non-edge cell phenotype resulted from the transcript increase of *PpCSP1-Citrine*. On the other hand, we could not find morphological and growth differences in protonemata and gametophores between wild-type and nPpCSP1-Citrine-nosT lines (Supplementary Fig. 8).

To investigate the relationship between *PpCSP1* and other factors involved in the reprogramming, we analysed transcript levels of *WOX13-like* genes[11] in nPpCSP1-Citrine-3′-UTR and nPpCSP1-Citrine-nosT lines with 5′-DGE during the reprogramming of cut leaves. However, no significant differences were observed in the transcript levels (Supplementary Fig. 6a,b). On the other hand, *PpCSP1* transcript levels investigated with the 5′-DGE data in Δ*ppwox13lab* line[11] were detected to be lower than those in wild type at 24 h after dissection, while *PpCSP1* transcripts were similarly induced until 6 h in wild type and the mutant (Supplementary Fig. 6c). These results suggest that *PpCSP1* is positively regulated by *WOX13-like* genes but *PpCSP1* does not regulate *WOX13-like* genes.

**PpCSP quadruple deletion attenuates reprogramming.** Deletion of the *PpCSP1* gene (Supplementary Fig. 7a) resulted in no detectable difference in reprogramming between wild type and the mutant (Fig. 5a,b). There are three closely related genes, *PpCSP2*, *PpCSP3* and *PpCSP4*, (Fig. 1d) in the *Physcomitrella* genome[39,51]. We generated single (*ppcsp2*, *ppcsp3* and *ppcsp4*), double (*ppcsp1* and *ppcsp2*), triple (*ppcsp1*, *ppcsp2* and *ppcsp3*) and quadruple (*ppcsp1*, *ppcsp2*, *ppcsp3* and *ppcsp4*) deletion mutants (Supplementary Fig. 7a–g). The percentage of excised leaves with reprogrammed cells was similar to the wild type in all single-, double- and triple-deletion mutant lines in both edge and non-edge cells (Fig. 5a,b). However, in the quadruple deletion mutant lines, cell protrusion was delayed (Fig. 5c). The delay was more severe in non-edge cells and was significant until 72 h (Fig. 5c,d), when chloronemata covered the excised leaves and further observation was impossible. Collectively, these results indicate that the four *PpCSP* genes are positive regulators of the reprogramming and possess redundant functionality.

PpCSP1 was expressed in not only stem cells but also proliferating non-stem cells in gametophore apices (Supplementary Fig. 1e) and appeared to localize at the phragmoplast (Supplementary Fig. 3 and Supplementary Movie 2). These data suggest the possibility that PpCSP1 is not involved in the reprogramming but in general cell cycle progression. To examine this possibility, we analysed the phenotype of the quadruple deletion mutant and the *PpCSP1* transcript-increased line in protonemata and gametophores. We could not distinguish the protonemata and gametophores of the quadruple deletion mutant and the transcript-increased line from those of wild type (Supplementary Fig. 8a–f). Moreover, the duration of cell cycles

of protonemata of these lines was measured with time-lapse observation and we could not find any differences (Supplementary Fig. 8g and Supplementary Movie 6). These results suggest that PpCSP1 does not play a major role in cell cycle progression in protonemata.

When we added a DNA synthesis inhibitor, aphidicolin to cut leaves, cell cycle re-entry was arrested but leaf edge cells still protruded, indicating that cell cycle progression is not required for reprogramming[9] (Supplementary Fig. 9). To examine whether PpCSP1 regulates reprogramming regardless of cell cycle, we treated with aphidicolin the quadruple deletion mutant, *PpCSP1* transcript-increased line, and wild type, and compared their reprogramming phenotype. In the presence of aphidicolin, the *ppcsp* quadruple deletion mutant and the *PpCSP1* transcript-increased line exhibited attenuated and enhanced reprogramming, respectively as in the absence of the cell cycle inhibitor (Supplementary Fig. 9). These indicate that PpCSP1 functions in reprogramming independent of cell cycle progression.

**Discussion**

On the basis of the results of this study, we propose a model for the function of *PpCSP1* in the cellular reprogramming of *Physcomitrella* (Fig. 5e). *PpCSP1* mRNA is weakly transcribed and degraded through regulatory elements localized to the 3′-UTR in all leaf cells (Fig. 3b–g). Subsequent to excision, a wound signal induces promoter activity, which results in an increase in transcript and protein levels (Fig. 2d–f). The increase of promoter activity is strong enough for reprogramming in edge cells but not in non-edge cells. Since some edge cells are not reprogrammed (Fig. 2a,d), another unidentified factor (X) must be necessary for uniform edge cell reprogramming. Furthermore, since some reprogramming still occurs in the *ppcsp* quadruple deletion line (Fig. 5c), another inductive pathway occurring independent of PpCSP1 must exist (Fig. 5e). In the nPpCSP1-Citrine-nosT and PpCSP1pro-PpCSP1-Citrine lines, without repression mediated by the 3′-UTR, PpCSP1 expression increases and triggers reprogramming in non-edge cells (Fig. 4a–g).

Shared domain structures and amino acid similarities between PpCSP1 and Lin28 (Fig. 1c,d) suggest that Lin28 is the most closely related protein of PpCSP1 in the metazoan genomes. Both *PpCSP1* and *Lin28* are dispensable for reprogramming and function to enhance the reprogramming. *Lin28* is dispensable for iPSC formation and promotes the maturation of iPSCs[12,13,52], although *Lin28* participates in iPSC reprogramming from human fibroblast cells[14]. In the *ppcsp* quadruple deletion line of *Physcomitrella*, reprogramming was attenuated in edge cells but was not completely arrested (Fig. 5c,d). Non-edge cells were effectively reprogrammed in the *PpCSP1* transcript-increased lines (Fig. 4b,f). However, the molecular mechanisms underlying *PpCSP1* and *Lin28* regulation appear to be different. Lin28 binds to precursors of miRNA *let-7* and inhibits its processing[31,32,34],

**Figure 3 | 3′-UTR of *PpCSP1* gene has a universal degradation function.** (**a**) Location of 5′-end of *PpCSP1* and *PpCSP1-Citrine* transcripts in wild-type, nPpCSP1-Citrine-3′-UTR and nPpCSP1-Citrine-nosT lines, respectively, detected by 5′-DGE transcriptome analysis. Sequence reads of full-length mRNAs were mapped around the transcription start site of the gene, and those of degraded mRNAs were mapped to other region of the transcript. (**b,d**) Bright-field (BF) and sGFP images of an excised leaf of EF1αpro:sGFP-nosT #3 (**b**) and EF1αpro:sGFP-3′-UTR #18 (**d**) at 0, 24 and 48 h after cutting. Several edge and non-edge cells are numbered for **c,e**, respectively. (**c,e**) The intensity of the sGFP signals in each cell of an excised leaf of EF1αpro:sGFP-nosT #3 (**c**) and EF1αpro:sGFP-3′-UTR #18 (**e**) (numbers correspond to cells in the top panels of **b,d**, respectively). Red and black lines indicate the sGFP intensity in cells that were and were not reprogrammed into stem cells, respectively. (**f,g**) Bright-field (BF) and fluorescent images of EF1αpro:sGFP-nosT #3 (**f**) and EF1αpro:sGFP-3′-UTR #18 (**g**) in protonemata and a gametophore, respectively. (**h**) Schematic representation of the introduced fragments. Series of the *PpCSP1* 3′-UTR with different lengths (yellow boxes) were connected to *sGFP* (green arrows), which is constitutively expressed by the rice *Actin 1* promoter (orange arrows). These deletion constructs were introduced into *Physcomitrella* leaf cells with mRFP (red arrow) fragments (shown at the top) by particle bombardment. (**i**) Representative cells with mRFP (red) and sGFP (green) signals with constructs shown in **h**. (**j**) Ratio of sGFP intensity to co-transformed mRFP intensity in each transformed cell ($n = 10$). Error bars represent s.d. Scale bars, 50 μm (**b,d,i**); 500 μm (**f,g**).

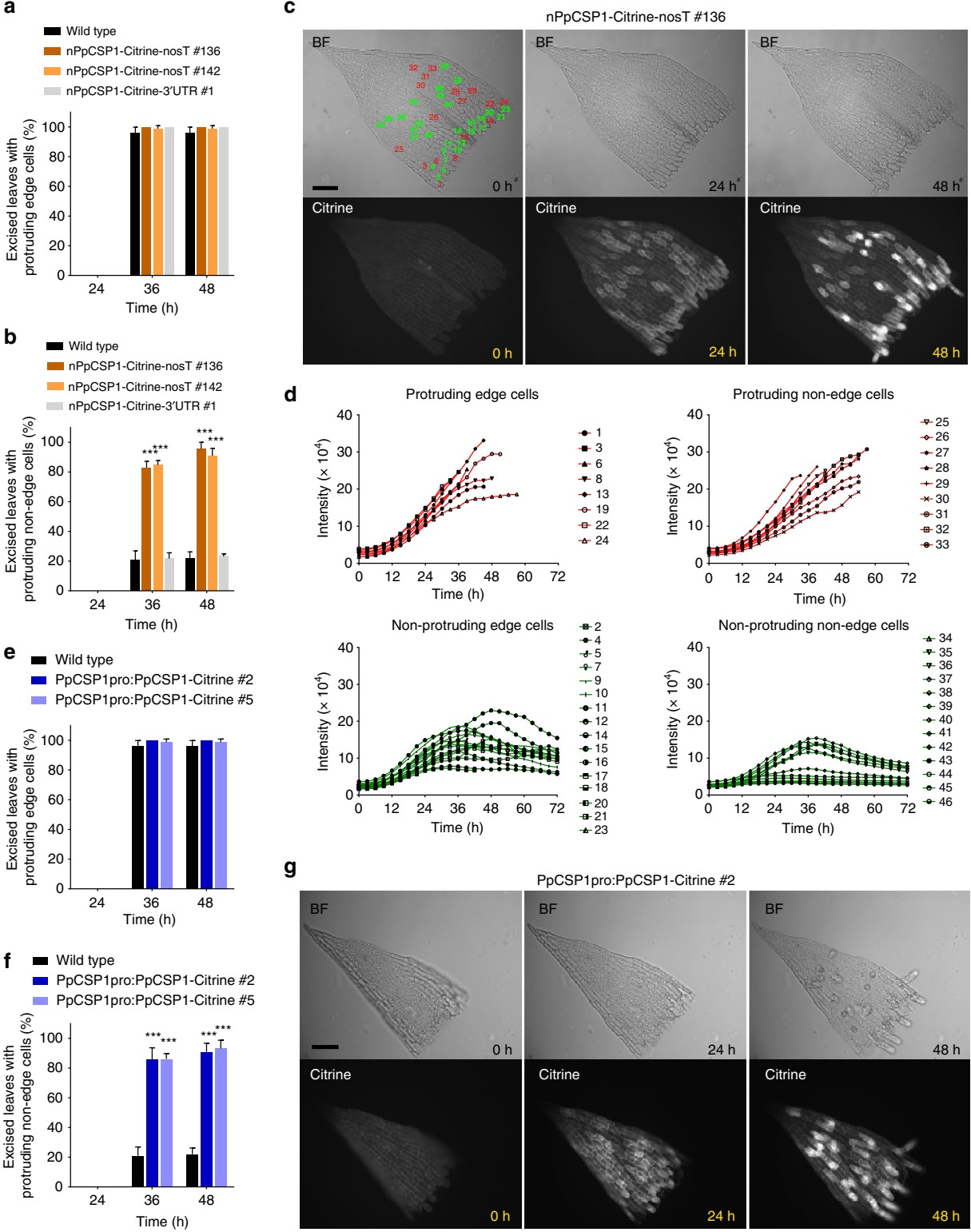

**Figure 4 | Increased PpCSP1 protein accumulation causes enhanced reprogramming.** (**a,b**) Percentages of excised leaves with protruding edge cells (**a**) and protruding non-edge cells (**b**). Twenty leaves excised from wild type, nPpCSP1-Citrine-nosT (#136 and #142), and nPpCSP1-Citrine-3'-UTR #1 were used for each analysis. Error bars represent s.d. from biological triplicates. ***$P < 0.001$ by two-sided Welch's $t$-test. (**c**) Expression pattern of PpCSP1-Citrine in an excised leaf of nPpCSP1-Citrine-nosT #136. Bright-field (BF) and Citrine images at 0, 24 and 48 h after cutting are shown. All edge cells and several non-edge cells are numbered for **d**. See also Supplementary Movie 5. (**d**) The intensity of Citrine signals in each cell (numbers correspond to cells in the top panel of **c**) of an excised leaf of nPpCSP1-Citrine-nosT #136. Red and green lines indicate the intensities of Citrine signals in protruding and non-protruding cells, respectively. (**e,f**) Percentage of excised leaves with protruding edge cells (**e**) and protruding non-edge cells (**f**). Twenty leaves were excised from wild type and PpCSP1pro:PpCSP1-Citrine (#2 and #5). Error bars represent the s.d. from biological triplicates. ***$P < 0.001$ by two-sided Welch's $t$-test. (**g**) Expression patterns of PpCSP1-Citrine in an excised leaf of PpCSP1pro:PpCSP1-Citrine #2. BF and Citrine images at 0, 24 and 48 h after cutting are shown. Scale bars, 100 μm (**c,g**).

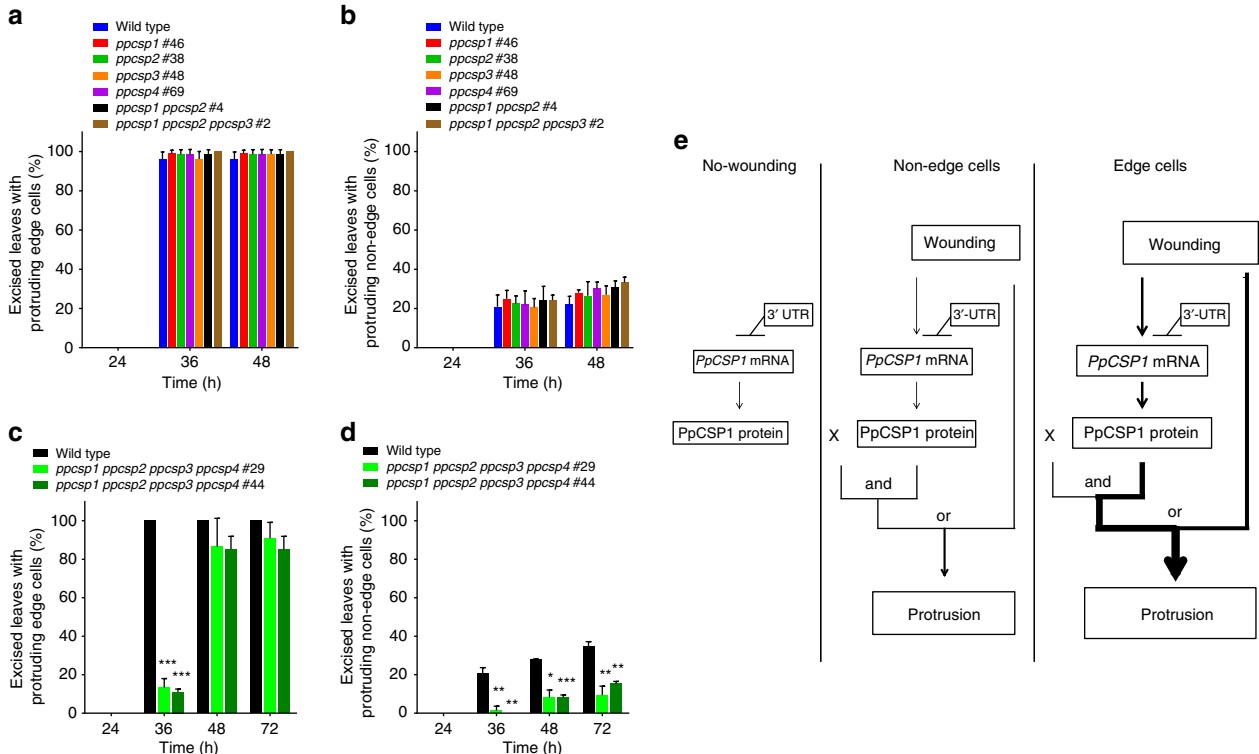

**Figure 5 | Inhibition of reprogramming in quadruple deletion mutants.** (**a,b**) Percentage of excised leaves with protruding edge cells (**a**) and protruding non-edge cells (**b**) in wild type, *ppcsp1* #46, *ppcsp2* #38, *ppcsp3* #48, *ppcsp4* #69, *ppcsp1 ppcsp2* #4 and *ppcsp1 ppcsp2 ppcsp3* #2. Twenty leaves were excised from each line. Error bars represent s.d. of biological triplicates. (**c,d**) Percentage of excised leaves of wild-type and *ppcsp1 ppcsp2 ppcsp3 ppcsp4* (#29 and #44) with tip growth from edge (**c**) and non-edge cells (**d**), respectively. Twenty leaves were excised from each line. Error bars represent s.d. of biological triplicates. *$P < 0.05$, **$P < 0.01$ and ***$P < 0.001$ by two-sided Welch's *t*-test. (**e**) Hypothetical model of the function of PpCSP1 in the reprogramming. The 3'-UTR represses PpCSP1 expression in both edge and non-edge cells. Signals from wounding are capable of overriding the repression and of effectively increasing PpCSP1 expression, resulting in activation of the reprogramming process.

while *let-7* leads to the degradation of *Lin28* transcripts[31]. Therefore, this negative feedback loop functions as a bistable switch to regulate cell fate[31]. We found that regulation of *PpCSP1* transcripts is mediated by its 3'-UTR but we could not find miRNA binding sites in this region nor *let-7* homologues in the *Physcomitrella* genome. Furthermore, the degradation of *PpCSP1* transcripts is not specific to the differentiated cells (Fig. 3b–e). The activation of the *PpCSP1* promoter in the reprogramming cells results in the increase of *PpCSP1* transcripts (Fig. 5e).

Multicellularity with stem cells has evolved independently in land plant and metazoan lineages and the molecular mechanisms underlying reprogramming appear to differ between these lineages[1–4]. Nevertheless, this study showed that closely related genes encoding CSD proteins, PpCSP1 and Lin28, are involved in reprogramming, although their orthology was not clear (Fig. 1d). Therefore, it is an open question whether PpCSP1 and Lin28 have evolved from a common gene or different genes of the last common ancestor.

CSD is highly conserved in bacteria, land plants and metazoa[19,20], but the biochemical function of CSD in reprogramming is unknown. In *Escherichia coli*, CSPs function as RNA chaperones that destabilize secondary structures in RNA[53,54] and deletion of four CSP genes results in growth defect under low temperature[53,54]. Wheat cold-shock domain protein 1 (WCSP1) also has nucleic acid binding activity, anti-termination activity and dsDNA melting activity[18]. Ectopic expression of WCSP1 in an *E. coli* CSP deletion mutant could complement its cold-sensitive phenotype[18], suggesting that the CSP function as RNA chaperone in response to cold stress is the ancestral function of CSP between bacteria and land plants. *Arabidopsis*

CSPs (AtCSPs) also function in the stress response and during regular development[17,55–60]. However, no report has shown that CSPs function in stem cell establishment/maintenance or reprograming in flowering plants. GUS reporter analysis showed that AtCSPs are expressed in shoot and root meristem harbouring stem cells[17,58–60]. These suggest that AtCSPs may play a role in stem cell regulation in *Arabidopsis*. It will be a future challenge to investigate the biochemical functions of CSD within PpCSPs and AtCSPs in reprogramming.

PpCSP1-Citrine signals localized at the phragmoplast when the reprogrammed leaf cells divide (Supplementary Fig. 3 and Supplementary Movie 2). The signals were maintained in the reprogrammed chloronema apical stem cells and diminished in the successive cell divisions, although the diminished signals were maintained in chloronema apical stem cells (Supplementary Fig. 3 and Supplementary Movie 2). In addition, PpCSP1 was expressed in both stem cells and proliferating non-stem cells in gameto-phore apices (Supplementary Fig. 1e). These results suggest that PpCSP1 is involved in cell cycle regulation during or after reprogramming, as Lin28 promotes cell cycle regulators and coordinates proliferative growth[26,27]. However, increasing and decreasing PpCSP1 levels in nPpCSP1-Citrine-nosT and the quadruple deletion mutant lines, respectively did not change the duration of cell cycles in protonema apical stem cells (Supplementary Fig. 8). Moreover, aphidicolin blocks cell cycle re-entry, nevertheless cells facing the cut protruded without dividing, indicating that the reprogramming does not require cell cycle progression. In the presence of aphidicolin, the *PpCSPs* quadruple deletion mutant and *PpCSP1* transcript-increased line exhibited attenuated and enhanced reprogramming,

respectively (Supplementary Fig. 9). These results indicate that PpCSP1 plays a role in reprogramming. It is a future question whether PpCSP1 functions in cell cycle regulation during the reprogramming.

In human cells, overexpression of Lin28 with a set of pluripotency-associated transcription factors Oct4, Sox2 and Nanog enhances reprogramming of fibroblast cells into iPSCs[14]. In addition to *let-7*, Lin28 binds to various mRNAs including ~50% of the human transcripts with motifs of GGAG or GGAG-like, although it is still unclear how its global mRNA-binding ability contributes to iPSC reprogramming[61–63]. Future studies are warranted to investigate both the *PpCSP1* and Lin28 regulatory networks in order to find molecular mechanisms underlying the common positive reprogramming functions between PpCSP1 and Lin28.

## Methods

**Plant material.** The Gransden 2004 strain of *P. patens*[51] was used as the wild-type strain and cultured on BCDAT medium under continuous white light at 25°C (ref. 24). The third or fourth leaves were excised from gametophores 3 weeks after inoculation and put into liquid BCDAT medium to induce the reprogramming[9]. Polyethylene glycol-mediated transformation[24] was performed using 10 μg of linearized plasmid as below: protoplasts were prepared from 3-day-cultured protonemata which were incubated in 25 ml of 8% mannitol solution with 0.5 g Driserase (Kyowa Hakko Kogyo Co., Ltd) at 25 °C for 30 min. After filtrating the protonemata with 50-μm nylon mesh, the protoplasts were collected by centrifugation at 180*g* for 2 min at room temperature, and resuspended into 40 ml of 8% (w/v) mannitol. Centrifugation and washing steps were repeated twice. Washed protoplasts were suspended in MMM solution (8.3% mannitol, 0.1% MES-KOH (pH 5.6), and 15 mM MgCl$_2$) at $1.6 \times 10^6$ cells ml$^{-1}$. Then, 300 μl of the protoplast suspension and 300 μl of PEG/T solution (28.5% polyethylene glycol 6,000 in 7.2% mannitol, 0.1 M CaCl$_2$ and 10 mM Tris-HCl (pH 8.0)) were added into 30 μl of linearized plasmids. The protoplasts were incubated at 45 °C for 5 min, and then at 20 °C for 10 min in water baths. The transformed protoplasts were diluted to 8 ml with protoplast liquid culture medium (5 mM Ca(NO$_3$)$_2$, 1 mM MgSO$_4$, 45 μM FeSO$_4$, 0.18 mM KH$_2$PO$_4$ (adjusted to pH 6.5 with KOH), the alternative TES, 50 mg l$^{-1}$ ammonium tartrate, 6.6% mannitol and 0.5% glucose), poured into a 6-cm Petri dish, and kept under the dark condition at 25 °C overnight. The protoplasts were collected by centrifugation at 180*g* for 2 min at room temperature, and suspended in 8 ml of top layer protoplast regeneration medium (BCD medium supplemented with 5 mM ammonium tartrate, 10 mM CaCl$_2$, 0.8% agar and 8% mannitol) preheated at 45 °C. The suspended protoplasts were poured into four 9-cm dishes that contained solidified bottom layer of protoplast regeneration medium (BCD medium supplemented with 5 mM ammonium tartrate, 10 mM CaCl$_2$, 0.8% agar and 6% mannitol) which was covered with cellophane. After 3-day incubation under continuous light, the regenerating protoplasts were transferred to BCDAT medium containing antibiotics for selection for 2 weeks. Then, the plants were transferred to BCDATG medium, incubated for 1 week, and re-inoculated onto the selection medium again. Stable transformants were further analysed by PCR and DNA gel blot analyses.

**Accession numbers.** Sequence data of *PpCSP*s can be found from Phytozome *P. patens* V3.3 (DOE-JGI, http://phytozome.jgi.doe.gov/) under the following accession numbers: *PpCSP1* (Pp3c5_6070); *PpCSP2* (Pp3c6_23240); *PpCSP3* (Pp3c5_7920); and *PpCSP4* (Pp3c5_7880).

**Plasmid construction for expression analysis.** Primers used for plasmid construction are provided in Supplementary Table 1. To insert the CDS[64] in frame with the *PpCSP1* CDS, a *PpCSP1* genomic DNA fragment just before the stop codon and a fragment just after the stop codon, were amplified and cloned into pCTRN-NPTII 2 (AB697058); thereby generating nPpCSP1-Citrine-nosT line (Supplementary Fig. 1a). One microgram of circular Cre recombinase[25] expression plasmid (AB542060), as extracted from the *E. coli* DH5α strain with Wizard Plus SV Minipreps DNA Purification System kit (Promega) without any restriction enzyme digestion, was introduced into the PpCSP1-Citrine line to excise the selection marker cassette and the nopaline synthase terminator flanked by two loxP sites to generate the nPpCSP1-Citrine-3′-UTR line. The regenerated lines were screened not to grow on a medium containing 20 mg l$^{-1}$ G418 and candidate lines were further confirmed by PCR.

For the promoter reporter lines, a 2.2 kb fragment containing a gateway rfcA cassette (Invitrogen) and a terminator sequence of pea (*Pisum sativum*) rbcS3A gene was amplified by PCR from the plasmid pT1OG (LC126301) with the primer pair shown in Supplementary Table 1 and then transferred into the XbaI-HindIII cut pPIG1b-NGGII plasmid (AB537478), resulting in the plasmid pAK101. A luciferase-coding sequence was amplified from pGL4.10 (Promega) and inserted

into the *Stu*I site of pAK101, resulting in a gateway-luciferase binary vector pAK102. A 1.8 kb PpCSP1 promoter fragment was amplified and cloned into the pENTR/D-TOPO vector (Invitrogen). The PpCSP1pro:LUC plasmid was constructed by LR reaction between the entry plasmid and pAK102. This construct was introduced to nPpCSP1-Citrine-3′-UTR line to generate the PpCSP1pro:LUC nPpCSP1-Citrine-3′-UTR line (Supplementary Fig. 2c).

**Plasmid constructions of *PpCSP1* 3′-UTR deletion series.** Primers used for plasmid construction are provided in Supplementary Table 1. sGFP and mRFP were inserted into pTKM1 vector[30] to generate pTKM1-sGFP and pTKM1-mRFP vectors. Different lengths of the *PpCSP1* 3′-UTR were amplified with wild type genomic DNA as a template and inserted just after the sGFP coding sequence at the *Apa*I site (Fig. 3h).

**Transient expression using particle bombardment.** Sixty mg gold particles (1.6 μm diameter) were coated with equal quantities of each pair of pTKM1-mRFP/ pTKM1-sGFP plasmid DNA and bombarded by PDS-1000 (Bio-rad) under 94.5 KPa vacuum condition into 5-week-old gametophores. Digital images were obtained using an Olympus DP71 camera on a fluorescence microscope (SZX16, Olympus, Japan). Fluorescence intensity of specific leaf cells was quantified by ImageJ 1.48v.

**Plasmid construction for EF1αpro:sGFP-3′-UTR line.** Primers used for plasmid construction are given in Supplementary Table 1. Fragments of sGFP and sGFP-3′-UTR were amplified from pTKM1-sGFP-3′-UTR plasmid and cloned into pENTR/D-TOPO (Invitrogen) and subsequently inserted into the pT1OG vector (LC126301)[40] (Supplementary Fig. 4a,b).

**Plasmid construction for the PpCSP1pro:PpCSP1-Citrine line.** Primers used for plasmid construction are provided in Supplementary Table 1. A fragment of 2.1 kb *PpCSP1* promoter and *PpCSP1*-coding sequence was amplified from wild-type genomic DNA and inserted into pCTRN-NPTII with *Xho*I and *Bsr*GI sites. The fragment containing the *PpCSP1* promoter, PpCSP1-Citrine fusion gene and nptII expression cassette was subsequently digested by *Sma*I and inserted into the pPTA1 vector (LC122350) (which contains the targeting sequence to PTA1 locus[40]) to generate the PpCSP1pro:PpCSP1-Citrine line (Supplementary Fig. 5e).

**Plasmid construction for the deletion of *PpCSP* genes.** Primers used for plasmid construction are provided in Supplementary Table 1. To delete *PpCSP1*, *PpCSP2*, *PpCSP3* and *PpCSP4* in wild type *Physcomitrella*, genomic fragments containing the 5′- and 3′-flanking regions of each gene were inserted into the 5′-end and 3′-region of the nptII expression cassette of pTN182 (AB267706), of the hygromycin resistance cassette of pTN186 (AB542059), of the BSD expression cassette of p35S-loxP-BSD (AB537973) and of the Zeocin resistance cassette of p35S-loxP-Zeo (AB540628) plasmids, respectively. The generated constructs were digested by suitable restriction enzymes for gene targeting (Supplementary Fig. 7a–d).

To generate *ppcsp* quadruple deletion mutants, the *PpCSP1*-deletion construct was introduced into wild-type *Physcomitrella* to generate *ppcsp1* lines. The *PpCSP2*-deletion construct, *PpCSP3*-deletion construct, and subsequently *PpCSP4*-deletion construct were introduced into the *ppcsp1* lines to generate the *ppcsp1 ppcsp2* double-deletion mutants, *ppcsp1*, *ppcsp2* and *ppcsp3* triple-deletion mutants and *ppcsp1*, *ppcsp2*, *ppcsp3* and *ppcsp4* quadruple deletion mutants, respectively.

**DNA gel blot analysis.** DNA gel blot analysis[9] was performed as below: ~3 μg of genomic DNA was digested with appropriate restriction enzyme(s) (see Supplementary Figs 2, 4 and 7), run on 0.7% (w/v) SeaKemGTG agarose (BME, Rockland, ME, USA), and transferred to a Hybond N$^+$ nylon membrane (GE Healthcare, Chicago, IL, USA). Probe labelling, hybridization and detection were performed using the AlkPhos direct labelling and detection system with CDP-Star (GE Healthcare) according to the supplier's instructions. Primers used for probe amplification are provided in Supplementary Table 1.

**Phylogenetic analysis.** Phylogenetic analysis with Neighbor-Joining method[65] was performed with updated datasets[66] including sequences from *Klebsormidium flaccidum*[67]. The nr data set used was as of 17 Jan 2015.

BLASTP search against a data set consisting of the nr as of Jan, 2015, *Klebsormidium* data set from http://www.plantmorphogenesis.bio.titech.ac.jp/ ~algae_genome_project/klebsormidium/kf_download.htm *Pinus taeda* assembly 1.01 annotation v2 http://dendrome.ucdavis.edu/ftp/Genome_Data/genome/ pinerefseq/Pita/v1.01/Pita_Annotation_v2/, and *P. patens* v1.6 data set https:// www.cosmoss.org/physcome_project/linked_stuff/Annotation/V1.6/P.patens. V6_filtered_cosmoss_proteins.fas.gz, were performed using PpCSP1 through http://moss.nibb.ac.jp/cgi-bin/blast-nr-Kfl. According to BLASTP search, we noticed that Lin28 proteins are most similar to PpCSP1 in metazoan genomes. To

see whether PpCSP1 and other plant CSPs are most similar to Lin28 in land plant genomes, *C. elegans* Lin28 was used as a query and PpCSPs together with other plant CSPs were found. Top 700 for PpCSP1 and 600 for Lin28 hit sequences were recovered and aligned using MAFFT[68] with the einsi option through http://moss.nibb.ac.jp/cgi-bin/selectNalign and a preliminary tree was drawn with http://moss.nibb.ac.jp/cgi-bin/makenjtree. From both trees, sparse samplings of terminal taxa were performed to include human and mouse Lin28 homologues and *Amborella trichopoda*, *Arabidopsis* and *Physcomitrella* CSP homologues. These sequences were further aligned through http://moss.nibb.ac.jp/cgi-bin/selectNalign. Sites aligned ambiguously or having gaps were marked as excluded for further analysis using MacClade ver 4 (ref. 69). After removing proteins lacking conserved zinc-finger domains and choosing proteins with one cold-shock domain and two zinc-finger domains, the nexus file was submitted from http://moss.nibb.ac.jp/cgi-bin/makemltree1000. This selects an amino acid substitution model based on the data and performs maximum likelihood analysis using RAxML version 8.1.16. Bootstrap analysis was performed with 1,000 replicates prepared with SEQBOOT in PHYLIP and consensus was calculated with CONSENSE[70].

**RNA preparation and RT-qPCR analysis.** Total RNA was purified from protonemata and cut leaves with the RNeasy Micro Kit (Qiagen). First-strand cDNA was synthesized using the ReverTra Ace qPCR RT Master Mix (TOYOBO). RT-qPCR was performed using an ABI PRISM 7500 (Applied Biosystems) with the QuantiTect SYBR Green PCR Kit (Qiagen). The cycle conditions were: 50 °C for 2 min and 95 °C for 10 min as pre-treatments, 95 °C for 15 s and 60 °C for 1 min at 40 cycles as amplification. After amplification cycles, we carried out dissociation analyses for confirmation of target validity. The sequences of primers for RT-qPCR are listed in Supplementary Table 1. Standard curves were estimated by dilution series (1, 0.1, 0.01, 0.001 and 0.0001) of one wild-type cDNA sample. Each transcript level determined by RT-qPCR analysis was normalized with *TUA1*[9].

**Digital gene expression profiling with 5′-DGE analysis.** Transcriptome analyses with 5′-DGE analysis[10] were performed as below (DRA accession number DRR055536-DRR055559): From 5 to 10 μg of total RNA, poly(A)+ RNA was enriched with the FastTrack Kit (Thermo Fisher Scientific, Waltham, MA, USA). Then, first-strand cDNA was synthesized using biotin-labelled dT20 primers containing an *Eco*P15I site (Biotin-TEG-5′-CTATCAGCAGTTTTTTTTTTTTTTT TTTTT-3′) using PrimeScript II reverse transcriptase (Takara Bio, Shiga, Japan). DNA synthesis extended after the 5′-end of the mRNA, complementary to the biotin-labelled P2 DNA-RNA chimeric oligonucleotide containing an *Eco*P15I site and a GGG ribonucleotide sequence (Biotin-TEG-5′-CTGCCCCGGGTTCCTCAT TCTCTCAGCArGrGrG-3′). The second-strand cDNA was synthesized based on the P2 sequence. After digestion with *Eco*P15I, the fragments were captured with streptavidin beads and ligated with P1 adaptors which were produced by annealing P1-A (5′-CCACTACGGCCTCCGCTTTCCTCTCTATGGGCAGTCGGTGAT-3′) and P1-B-NN oligonucleotides (5′-N*N*ATCACCGACTGCCCATAGAGAGGA AAGCGGAGGCGTAGTGG-3′, where asterisks indicate phosphorothioate bonds). The resulting 25-bp 5′-cDNA fragments were amplified by 12 cycles of PCR using P1 (5-CCACTACGGCCTCCGCTTTCCTCTCTAT-3′) and P2 primers (5′-CTGCC CCGGGTTCCTCATTCT-3′) and were then subject to 25-bp SOLiD single-read sequencing from the P1 sites. For the comparison between nPpCSP1-Citrine-nosT and nPpCSP1-Citrine-3′-UTR lines, special reference for each line with the targeted change on the scaffold_41 where the *PpCSP1* locus is present was prepared and the sequence tags were mapped on the respective reference. Expression profiles of gametophore leaves 0, 1, 3 and 6 h after excision in nPpCSP1-Citrine-nosT line, nPpCSP1-Citrine-3′-UTR lines and wild type[11] were analysed. Cumulative sum of tags of *PpCSP1* transcript for all time points in these lines were calculated and shown in Fig. 3a.

**Microscopy and image analyses.** Live-imaging analysis was performed using a fluorescence microscope (IX81, Olympus) with a cooled CCD camera (ORCA-AG, Hamamatsu Photonics) or an EM-CCD camera (ImagEM, Hamamatsu Photonics). Protonemata were cultured on glass-bottom dishes with BCDAT medium for 5–7 days before time-lapse observation of protonema growth. For luciferase bioluminescence imaging, tissues were pre-cultured for 18 h in BCDAT medium, including 500 μM beetle luciferin potassium salt (Promega), before the observation. The third or fourth leaves were excised from gametophores on a plate 3 weeks after inoculation and placed on a 35-mm glass-based dish (IWAKI) covered with 2% methylcellulose. The leaves were covered with cellophane and then with 0.8% solid BCDAT medium. The petri dish was set on the stage of an IX81 microscope. Bright-field and Citrine-fluorescence images (using the ×10 objective lens) of excised leaves were taken at 20-min intervals for 72 h after excision. A U-MNIBA3 filter (Olympus) was used for Citrine. For the bioluminescence imaging with the time-lapse observation, images were taken at 2-h intervals for 72 h after the excision. A U-MGFPHQ filter (Olympus) was used for the detection. Between imaging, the stage was moved in continuous white light conditions under control of the MetaMorph software (Molecular Devices). The area and intensity of the Citrine, LUC or sGFP signal in each cell were calculated. The average intensity at each time point was calculated as the intensity of the GFP signal divided by the area of the cell. The movie of the time-lapse images was edited with ImageJ 1.48v.

Images of PpCSP1-Citrine localization (Fig. 2c) were taken by an inverted microscope (IX81, Olympus) equipped with a spinning-disk unit (CSU21, Yokogawa) with a CMOS camera (ORCA-Flash 4.0, Hamamatsu Photonics). Bandpass filters (FF01-550/88-25, Semrock) for Citrine were used in the spinning-disk unit. Gametophore apex images (Supplementary Fig. 1e) were taken by a fluorescence microscope (BX51, Olympus) equipped with a colour camera (DS-Fi1c, Nikon). Citrine-fluorescence images were taken with U-MNIBA3 filter (Olympus). Aphidicolin treatment[9] was performed as excised leaves were put into BCDAT liquid medium containing aphidicolin at the concentration denoted in Supplementary Fig. 9 or mock (DMSO). The leaves 72 or 120 h after excision were stained in a solution containing 0.1% aniline blue and 0.1% $K_3PO_4$ (pH 12.5) to visualize newly synthesized cell plates. Fluorescent images were taken by a fluorescence microscope (BX51, Olympus) equipped with a colour camera (DS-Fi1c, Nikon) and with long-pass filter (U-MWU2, Olympus; Supplementary Fig. 9). Protonema and gametophore images of EF1αpro:sGFP-nosT and EF1αpro:sGFP-3′-UTR lines (Fig. 3f,g) and bombardment experiment images (Fig. 3i) were taken by a fluorescence microscope (SZX16, Olympus) equipped with a colour camera (DP71, Olympus). sGFP fluorescence was taken by GFPHQ filter (Olympus). mRFP fluorescence was taken by an RFP1 filter (Olympus) for excitation and 593/40 filter (Semrock) for emission. Fluorescence linearity of the colour camera DP71 was examined with fluorescence beads. Images showing a fluorescence intensity that fitted within the linear range were chosen for quantitative analyses.

**Data availability.** We declare that all data supporting the findings of this study are available within the manuscript and its Supplementary Files or are available from the corresponding authors upon request.

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

## Acknowledgements

We thank all members in the laboratory of M.H. and the ERATO Reprogramming Evolution Project for useful discussions and technical support; and Shih-Long Tu, Chunli Chen, Xiangdong Fu, Yan Sun and Changwei Shao for helpful discussions. We also thank the Functional Genomics Facility, Model Plant Research Facility, Spectrography and Bioimaging Facility, and the Data Integration and Analysis Facility of National Institute for Basic Biology, and Japan Advanced Plant Science Network for their technical support. This work was partially funded by MEXT KAKENHI grants to Y.T. and M.H., JSPS KAKENHI grants to M.H. and JST grant to M.H., Program for Advancing Strategic International Networks to Accelerate the Circulation of Talented Researchers to M.H. and P.N.B., the Center for the Promotion of Integrated Sciences of SOKENDAI, the Howard Hughes Medical Institute and the Gordon and Betty Moore Foundation (through Grant GBMF3405) and from the NIH (R01-GM043778) to P.N.B.

## Author contributions

C.L., Y.Sako, M.I., Y.Sato, Y.T. and M.H. conceived and designed the experiments. C.L., Y.Sako, Y.Sato, A.I., Y.H., K.T. and Y.K. performed the experiments. C.L., Y.Sako, A.I., K.T., T.N., M.I., Y.Sato, S.-H.W., T.M., Y.T., P.N.B. and M.H. analysed the data. M.K., Y.H., D.K., T.M., P.N.B., Y.Sato and M.H. contributed materials. C.L., Y.T., P.N.B. and M.H. wrote the manuscript. All authors reviewed the manuscript.

## Additional information

**Competing financial interests:** The authors declare no competing financial interests.

