## [Peer Review File · Nature Communications]

Reviewers' comments:

Reviewer #1 (Remarks to the Author):

The manuscript describes the analysis of the moss Cold-Shock-Domain Protein 1 (PpCSP1) gene that shares a conserved protein domain structure with the mammal Lin28 factor, a component that in human fibroblasts contributes to the acquisition of stem cell identity. The authors show that PpCSP1 is transcriptionally activated at the margins of gametophore leaf cuts prior to chloronema outgrowth and that the PpCSP1 promoter remains active in growing chloronema filaments in the apical cell, which is considered to have acquired stem cell identity. Similar to Lin28 mRNA PpCSP1 transcript stability is controlled by 3'UTR sequences although apparently not via a micro-RNA related mechanism. With respect to chloronema outgrowth increased PpCSP1 transcript levels positively relate to the number of outgrowing filaments and also affect non-marginal cells not subject to reprogramming in wild type. In contrast, only a simultaneous knockdown of PpCSP1 and 3 additional gene relatives reduces reprogramming and chloronema outgrowth, the PpCSP1 function in contrast to Lin28 thus is not unique.

The experimental data are of high quality and nicely presented; two minor points are listed below:

L233-54 The nomenclature of the 3' termini is hard to follow through the manuscript; replacement of the native 3'UTR by the NOS terminator might be immediately visible in the construct name for example in the knock-in constructs nPpCSP1-citrine-3'UTR /nPpCSP1-citrine-NOS

L 275-6 and L305-7 what is meant with „overcomes degradation"? Measured are steady state levels and the CSP1 transcript half-life-time probably differs between the native and the nos 3'UTR; increased promoter activity consequently should relate to higher steady state transcript levels.

A major concern relates to the phylogeny (Fig. 1d), firstly, without a convincing bootstrap value at the missing branch point the two paraphyletic groups of protein sequences provide no evidence for an „orthologous relationship" (L109-111).

Secondly, this phylogeny raises serious questions with respect to the general conclusion and the title that PpCSP1 and Lin28 in moss or mammals, respectively, identify an unifying mechanism for the gain of pluripotency. In contrast, the need for a quadruple mutant and the broad PpCSP1 response at the leaf cut in moss suggest that in difference to the Lin28 specificity in human fibroblasts, moss CSP genes provide a more general, basic competence and relating to chaperonins probably related to a stress response.

In my view the phylogeny would needs substantial improvement to allow the far-reaching suggestion about an ancestral reprogramming mechanism shared between animal stem cells and moss chloronema apical cells. Alternatively, an open and critical discussion would be essential that unbiasedly considers common ancestry or co-evolution, but also possible

differences between apical growth of chloronema filaments after wounding or animal stem cells.

Reviewer #2 (Remarks to the Author):

Revision of "Cold-shock domain proteins reprogram differentiated cells to stem cells in both land plants and mammals" by Li et al.

In this manuscript Li et al. characterize genes encoding Cold-Shock Domain Proteins (CSP) in *Physcomitrella patens*. They found that PpCSP1 is specifically induced in reprogramming cells. Furthermore, they show that an increase of PpCSP1 and or a decrease of PpCSP activity as seen in a quadruple mutant quantitatively enhance or reduce reprogramming, respectively. The authors further show that the PpCSP1 3' UTR affects mRNA stability and that this regulation is important to adjust PpCSP1 protein levels, although the precise mechanism could not be identified. As Lin28 in mammals and PpCSP1 share the domain structure, the authors suggest a general function for CSP proteins in reprogramming of differentiated cells.

I think that this manuscript provides a thorough description of the participation of PpCSP1 and other cold-shock proteins during reprogramming of differentiated cells in *Physcomitrella patens*, specially the studies on the PpCSP1 reporters and CSP quadruple knock outs seem very interesting.

I also think that the manuscript would be strengthened by a characterization of the molecular mechanisms underlying PpCSP1 regulation and/or action. For example, the relationship of PpCSP1 with other factors known to affect reprogramming of *Physcomitrella* cells, such as WOX13-like genes, is not explored.

Specific comments:

1) After reading the title, I expected that the authors would actually carry out experiments with animal genes. For example, expression of Lin28 or Lin28-PpCSP1 chimeras in *Physcomitrella*. Considering the actual data, I suggest to modify the title or include additional experiments with Lin28.

2) The current view suggests that plant and animal microRNAs evolved independently of each other. Therefore, I think that the section "PpCSP1 does not appear to be regulated by a microRNA", which begins with the search of a potential microRNA regulating PpCSP1 based on the regulation of Lin28 mRNA by let-7 miRNA is not sound as it is.

3) Are there additional phenotypic effects caused by the increase or decrease of CSP levels? Please describe in more detail.

4) I think that the concept of this sentence might be explained in a better way: "In edge cells that never protruded, LUC signals initially increased but decreased after reaching a

maximum."

5) I also found this sentence confusing: "The smaller variation in protein levels than in promoter activity in cells that eventually protrude suggests the potential involvement of post-transcriptional regulation of PpCSP1."

Reviewer #3 (Remarks to the Author):

Li et al.

Cold-shock domain proteins reprogram differentiated cells to stem cells in both land plants and mammals

Li et al report that members of the cold-shock domain gene family increase the efficiency of stem cell reprogramming in the moss *Physcomitrella*. As homologous, perhaps orthologous proteins perform similar roles on animals, such a role may reflect deep homology. The authors further show that, as is the case in metazoans, the stability/translatibility of the mRNA is a key regulatory point in the accumulation of the proteins. A key question is what are the targets for these proteins in both plants and animals and are they conserved as well?

General comments:

Given that miRNAs may have evolved independently in plants and animals (or perhaps this idea has been challenged?), it is not surprising that this aspect of the mode of regulation is not conserved.

The text is quite repetitive and can be reduced significantly without loss of information or take home message.

Reviewer #4 (Remarks to the Author):

This work reveals a possible parallel in the genetic control of plant and animal stem cells. The paper focuses on a moss homolog of Lin28, which is associated with stem cell identity in animals. Lin28 is expressed in multiple stem cell types and facilitates induction of iPSCs by Oct4, Sox2 and Nanog, but mouse knockouts showed that Lin28 is not essential for stem cell functions. Lin28 has been implicated in RNA processing, particularly in the maturation of the differentiation-promoting let-7 miRNA, but it also appears to regulate translation or stability of a large number of mRNAs.

Li et al found that the moss Lin28 homolog PpCPS1 is activated during the de-differentiation of leaf cells into chloronema apical cells, which formally function as stem cells that sustain

growth of the chloronema. The authors saw that PpCPS1 expression is negatively regulated by sequences present in the 3' region of the mRNA and took advantage of this to show that PpCPS1 overexpression increased the frequency of chloronema formation from cut leaves. Mutation of all four PpCPS1 paralogs delayed, but did not abolish emergence of chloronema from leaves.

The exact mechanism by which PpCPS1 or Lin28 affect cell differentiation and/or self-renewal remains unknown, so it is not clear at the moment whether this reflects a common mechanism in the control of stem cell functions across kingdoms. However, it is striking that a gene involved in re-directing cell fate in plants is the closest homolog of a gene known to facilitate de-differentiation in animals and therefore the paper should be of wide interest.

The paper is clearly written, the experiments are logical and the methods are properly described. However, the conclusion that PpCPS1 has a conserved role that is specific to stem cells remains open to criticism. I think the authors need to address the following points before publication:

1. In the Supplementary video 1, it appears that PpCPS1-Citrine is at first expressed in the emerging chloronemal cells, but during extension of the chloronema, expression is lost from the apical cell. This contrasts with the preferential expression in apical cells shown in Figure 1a,b. The video also gives the impression that PpCPS1 expression oscillates during cell division. Is PpCPS1 expression really associated with stem cell function or with cell cycle activation? Can the authors show examples of moss cells that divide actively but do not express PpCPS1, and of stem cells that are quiescent but still express PpCPS1?

2. If Lin28 has a role in promoting an undifferentiated state across kingdoms, it would be expected that the close homologs in other plants would also be involved de-differentiation and/or meristem functions. The closest homolog in Arabidopsis is CSP1, which has been implicated in mRNA processing during cold shock but so far not in stem cell functions. Have the authors investigated a possible role of Arabidopsis CSP1 in meristem function? This point should be at least discussed.

3. The assay for "cell reprogramming" is based primarily on the number of excised leaves with cells protruding from the cut edge or from the leaf blade. It would be more convincing to include additional criteria for reprogramming, for example, the number of cells expressing a marker gene for chloronema apical cell identity.

Minor points:

4. In Figures 1a-b, 2a and 3b,d it is necessary to include a negative controls showing lack of autofluorescence in untransformed cells.

5. In page 10: " The smaller variation in protein levels than in promoter activity in cells that eventually protrude suggests the potential involvement of post-transcriptional regulation of PpCSP1." The results could equally be explained by stability of the pPCSP1 protein.

Reply to Reviewers' comments:

Reviewer #1 (Remarks to the Author):

L233-54 The nomenclature of the 3' termini is hard to follow through the manuscript; replacement of the native 3'UTR by the NOS terminator might be immediately visible in the construct name for example in the knock-in constructs nPpCSP1-citrine-3'UTR /nPpCSP1-citrine-NOS

We changed from “nPpCSP1-Citrine” to “nPpCSP1-Citrine-nosT”.

L 275-6 and L305-7 what is meant with „overcomes degradation”? Measured are steady state levels and the CSP1 transcript half-life-time probably differs between the native and the nos 3'UTR; increased promoter activity consequently should relate to higher steady state transcript levels.

We agreed with the comment and removed “and overcomes the degradation” from line 344. For previous line 306, we changed from “The increase of *PpCSP1* transcripts depends on the activation of the *PpCSP1* promoter in the reprogramming cells, which overcomes the degradation activity (Fig. 5e)” to “The activation of the *PpCSP1* promoter in the reprogramming cells results in the increase of *PpCSP1* transcripts (Fig. 5e)” (line 381).

A major concern relates to the phylogeny (Fig. 1d), firstly, without a convincing bootstrap value at the missing branch point the two paraphyletic groups of protein sequences provide no evidence for an „orthologous relationship" (L109-111).

We noticed on this point and wrote in the previous manuscript that “The obtained phylogenetic tree was consistent with the orthologous relationship of Lin28 and related proteins in metazoa with plant CSPs including PpCSP1 (Fig. 1d)”. To further clarify, we changed the sentence to “Although it was not clear whether PpCSP1 is orthologous or paralogous to Lin28 (Fig. 1d), PpCSP1 and Lin28 should be homologous and these results led us to investigate whether PpCSP1 plays a role similar to Lin28 in reprogramming differentiated cells to stem cells.” (line 115)

Secondly, this phylogeny raises serious questions with respect to the general conclusion and the title that PpCSP1 and Lin28 in moss or mammals, respectively, identify an unifying

mechanism for the gain of pluripotency. In contrast, the need for a quadruple mutant and the broad PpCSP1 response at the leaf cut in moss suggest that in difference to the Lin28 specificity in human fibroblasts, moss CSP genes provide a more general, basic competence and relating to chaperonins probably related to a stress response. In my view the phylogeny would need substantial improvement to allow the far-reaching suggestion about an ancestral reprogramming mechanism shared between animal stem cells and moss chloronema apical cells. Alternatively, an open and critical discussion would be essential that unbiasedly considers common ancestry or co-evolution, but also possible differences between apical growth of chloronema filaments after wounding or animal stem cells.

The number of alignable amino acid residues between animal and plant CSD proteins is limited and it is not possible to improve the phylogenetic tree. Therefore, we follow the alternative option proposed by this reviewer to further discuss both similarity and difference between moss and mammal reprogramming. We mentioned both points as “This enhancement of reprogramming is similar to that of PpCSP1, although the regulatory mechanisms are different between Lin28 and PpCSP1” in the last paragraph of discussion in the previous version, and it is expanded to the last three paragraphs in Discussion of this version. We also changed the title to clarify PpCSP1 is a homolog to Lin28.

Reviewer #2 (Remarks to the Author):

I also think that the manuscript would be strengthened by a characterization of the molecular mechanisms underlying PpCSP1 regulation and/or action. For example, the relationship of PpCSP1 with other factors known to affect reprogramming of *Physcomitrella* cells, such as WOX13-like genes, is not explored.

We agreed this reviewer’s comment to strengthen this manuscript and analyzed the transcript levels of *PpWOX13-like* genes (Sakakibara et al. 2014 Development 141: 1660) in the 5’-DGE data using nPpCSP1-Citrine-3’UTR and nPpCSP1-Citrine-nosT lines (Supplementary Fig. 5a,b). We also analyzed *PpCSP1* transcript levels in the *ppwox13-like* deletion mutant line (Supplementary Fig. 5c). These analyses suggested positive regulation of *PpCSP1* by PpWOX13-like genes (line 307).

Specific comments:

1) After reading the title, I expected that the authors would actually carry out experiments

with animal genes. For example, expression of Lin28 or Lin28-PpCSP1 chimeras in *Physcomitrella*. Considering the actual data, I suggest to modify the title or include additional experiments with Lin28.

We followed the suggestion and changed the title as “A Lin28 homolog reprograms differentiated cells to stem cells in the moss *Physcomitrella patens*”.

2) The current view suggests that plant and animal microRNAs evolved independently of each other. Therefore, I think that the section "PpCSP1 does not appear to be regulated by a microRNA", which begins with the search of a potential microRNA regulating PpCSP1 based on the regulation of Lin28 mRNA by let-7 miRNA is not sound as it is.

We revised the description to stress the independent evolution of plant and animal microRNAs (line 232).

3) Are there additional phenotypic effects caused by the increase or decrease of CSP levels? Please describe in more detail.

We could not find additional phenotypic effects in both protonemata and gametophores and added the description in line 301 and line 336.

4) I think that the concept of this sentence might be explained in a better way: "In edge cells that never protruded, LUC signals initially increased but decreased after reaching a maximum."

We changed the sentence to “LUC signals initially increased but were not maintained as the protruded edge cells (Fig. 2e right)” (line 174).

5) I also found this sentence confusing: "The smaller variation in protein levels than in promoter activity in cells that eventually protrude suggests the potential involvement of post-transcriptional regulation of PpCSP1."

We revised the sentence to "The smaller variation in protein levels than in promoter activity in cells that eventually protrude (Fig. 2e, f; left) suggests the potential involvement of post-transcriptional regulation or the differences in stability of the transcripts and proteins of PpCSP1." (line 183).

Reviewer #3 (Remarks to the Author):

Given that miRNAs may have evolved independently in plants and animals (or perhaps this idea has been challenged?), it is not surprising that this aspect of the mode of regulation is not conserved.

There are similarities between animal and plant miRNAs as well as differences. We mentioned more carefully on this point (line 232).

The text is quite repetitive and can be reduced significantly without loss of information or take home message.

We deleted the following sentences to make the manuscript more concise.

In Results, “The obtained phylogenetic tree was consistent In each lineage.” was deleted (line 115).

In Results, “Given that *Lin28* mRNA..... to degrade the transcripts in *Physcomitrella*” was deleted (line 232).

In Results, “including miRNAs targeting *Lin28* such as *let-7*” was deleted (line 257).

In Discussion, “By BLAST searches to non-redundant two zinc-finger domains” was deleted (line 355).

Reviewer #4 (Remarks to the Author):

1. In the Supplementary video 1, it appears that PpCPS1-Citrine is at first expressed in the emerging chloronemal cells, but during extension of the chloronema, expression is lost from the apical cell. This contrasts with the preferential expression in apical cells shown in Figure 1a,b.

The decrease of intensities at the chloronema apical cell in supplementary movie 1 is caused by the away of the apical stem cell from the focal plane. We made a caution in the legend of Supplementary Video 1.

The video also gives the impression that PpCPS1 expression oscillates during cell division.

We could not detect oscillation of PpCSP1 expression. We added a new supplementary Video 2 to show stable PpCSP1 expression at protonema apical stem cells.

Is PpCPS1 expression really associated with stem cell function or with cell cycle activation? Can the authors show examples of moss cells that divide actively but do not express PpCPS1, and of stem cells that are quiescent but still express PpCPS1?

All stem cells divide and there are no quiescent stem cells in *Physcomitrella protonemata*. So it is not possible to do the requested experiment. Instead, we added that increase and decrease of PpCSP1 levels in nPpCSP1-Citrine-nosT and quadruple deletion mutant lines, respectively did not cause effects in cell cycle progression in protonema growth in line 301 and line 336.

2. If Lin28 has a role in promoting an undifferentiated state across kingdoms, it would be expected that the close homologs in other plants would also be involved de-differentiation and/or meristem functions. The closest homolog in Arabidopsis is CSP1, which has been implicated in mRNA processing during cold shock but so far not in stem cell functions. Have the authors investigated a possible role of Arabidopsis CSP1 in meristem function? This point should be at least discussed.

We have not examined PpCSP1 orthologs in Arabidopsis. Therefore, we discussed the possible role of Arabidopsis CSP1 in meristem function (line 406).

3. The assay for "cell reprogramming" is based primarily on the number of excised leaves with cells protruding from the cut edge or from the leaf blade. It would be more convincing to include additional criteria for reprogramming, for example, the number of cells expressing a marker gene for chloronema apical cell identity.

All protruded cells become chloronema apical cells (Ishikawa et al. 2011 Plant Cell 23: 2924) and "protrusion" is the most reliable marker to identify reprogramming at present. Although *PpWOX13-like* genes are induced during the reprogramming (Sakakibara et al. 2014 Development 141: 1660), the level of induction varies from cell to cell and sometimes

increases in non-reprogramming cells. To stress the reliability of “protrusion”, we added the following sentence (line 140).

“All leaf cells with tip growth behave as chloronema apical stem cells (Ishikawa et al. 2011) and the acquisition is the most reliable sign of the reprogramming at present.”

Minor points:

4. In Figures 1a-b, 2a and 3b,d it is necessary to include a negative controls showing lack of autofluorescence in untransformed cells.

Photos of wild type plants as a negative control are added in Supplementary Figure 1.

5. In page 10: " The smaller variation in protein levels than in promoter activity in cells that eventually protrude suggests the potential involvement of post-transcriptional regulation of PpCSP1." The results could equally be explained by stability of the pPCSP1 protein.

We agree and added to mention on the stability (line 185).

Reviewers' comments:

Reviewer #1 (Remarks to the Author):

The authors addressed the criticism raised for the previous version. However, some language editing still is required!

line 110 and 326-328: The ancestry is simply unclear due to long independent evolutionary trajectories; there is neither an argument against a common ancestor nor evidence for an ancient gene duplication suggested by „paralogous“

line 167/8: In the *Physcomitrella*

line 169: median length (334 bp) of 3' UTRs

line 206: miRNAs (are?) evolved

lines 206-228 why not PpCSP1 3' UTR throughout?

line 222: what is meant by "the same terminator was placed at the end" the authors deal with polyadenylation/mRNA stability not with termination of transcription and of course any secondary structure would be harmed by UTR deletions.

line 307/308: sentence

Reviewer #2 (Remarks to the Author):

I have previously expressed concern about the section "PpCSP1 does not appear to be regulated by a microRNA". -- The authors kept the section and added some comments.

Along these lines, I suggest to consider the following points:

Line 224, "We subsequently searched candidate miRNAs using the 3' UTR as a BLAST query in the miRBase website (www.mirbase.org). However, we could not find any miRNA targeting sequences in the 3' UTR." – Note that BLAST is not the right tool to perform a bioinformatic search for miRNA targets and discard a miRNA targeting.

There are experimental reports about genome-wide identification of miRNA targets in *Physcomitrella* using RNA-seq and degradome analysis (e.g., Addo-Quaye et al., RNA 2009). I think the authors should confirm that PpCSP1 does not appear as a potential miRNA target in the literature and mention that in the text.

Reviewer #4 (Remarks to the Author):

The main questions raised in my previous review were: 1) what is the nature of the "reprogramming" seen in moss, and 2) can this be clearly separated from a role of PpCSP1

in cell cycle progression. In the revised version, I do not think these questions have been addressed satisfactorily.

In relation to question 1:

Looking at Supplementary video 1, I am not convinced that the apparent decrease in the expression of PpCSP1 in some apical cells is due simply to the cells moving out of the focal plane. Based on the bright field images, some apical cells are clearly out of focus but show strong PpCSP1 signal, whereas others are closer to the focal plane but show little expression (e.g. the apical cell just about to cross the lower edge of the image at the end of the video). In addition, the video was cropped in a way that did not allow me to check whether the pattern shown for normal protonemata (Supplementary video 2) is quickly established during regeneration from cut leaves.

Although the authors state that they could not detect oscillation of PpCSP1 expression during the cell cycle, the videos showing PpCSP1 expression during regeneration from leaves (Supplementary videos 1, 3, 4) do show a striking pattern with expression peaking just before or during cytokinesis, after which some of the protein apparently remained associated with the newly made cell wall. This pattern looks different from that seen in protonemal cells (Supplementary video 2) and suggests a role associated with cell cycle progression.

In reply to the question whether PpCSP1 might play a role in cell cycle progression rather than specifically in stem cell identity, the authors state that "All stem cells divide and there are no quiescent stem cells in *Physcomitrella protonemata*. So it is not possible to do the requested experiment". The experiment requested does not need to be confined to protonemata. If the authors can show at any stage of the moss life cycle examples of dividing cells that do not express PpCSP1, this would be sufficient to break the correlation between PpCSP1 and cell cycle progression.

The authors also state that "Instead, we added that increase and decrease of PpCSP1 levels in nPpCSP1-Citrine-nosT and quadruple deletion mutant lines, respectively did not cause effects in cell cycle progression in protonema growth in line 301 and line 336". However, I could not find the new data showing that altered PpCSP1 levels did not affect cell cycle progression.

In relation to question 2:

Reprogramming normally refers to changes in gene expression that underlie a change in cell identity. For this reason, it would be important to show that changes in the expression of other stem cell markers are among the earliest effects of PpCSP1, before any visible changes in growth or cell cycle progression. If the only way to detect the re-establishment of stem cells in moss is through morphological changes, then the authors should not use the word re-programming in a way that suggests a process similar to the induction of iPSC by Oct4, Sox2 and Nanog in mammals.

Reply to Reviewers' comments:

Reviewer #1

line 110 and 326-328: The ancestry is simply unclear due to long independent evolutionary trajectories; there is neither an argument against a common ancestor nor evidence for an ancient gene duplication suggested by „paralogous“

We changed “Although it was not clear whether *PpCSP1* is orthologous or paralogous to *Lin28* (Fig. 1d)” to “Although the low resolution of the phylogenetic tree did not enable us to examine whether *PpCSP1* is orthologous or paralogous to *Lin28* (Fig. 1d)” in line 113-114 in this file.

line 167/8: In the *Physcomitrella*

line 169: median length (334 bp) of 3' UTRs

line 206: miRNAs (are?) evolved

lines 206-228 why not *PpCSP1* 3' UTR throughout?

We changed the wordings.

line 222: what is meant by "the same terminator was placed at the end" the authors deal with polyadenylation/mRNA stability not with termination of transcription and of course any secondary structure would be harmed by UTR deletions.

The sentence was deleted.

line 307/308: sentence

We changed “Both *PpCSP1* and *Lin28* are dispensable for reprogramming, functioning to enhance the reprogramming.” to “Both *PpCSP1* and *Lin28* are dispensable for reprogramming and function to enhance the reprogramming.” in line 353-354 in this file.

Reviewer #2

Line 224, “We subsequently searched candidate miRNAs using the 3’ UTR as a BLAST query in the miRBase website (www.mirbase.org). However, we could not find any miRNA targeting sequences in the 3’ UTR.” – Note that BLAST is not the right tool to perform a bioinformatic search for miRNA targets and discard a miRNA targeting.

There are experimental reports about genome-wide identification of miRNA targets in *Physcomitrella* using RNA-seq and degradome analysis (e.g., Addo-Quaye et al., RNA 2009). I think the authors should confirm that PpCSP1 does not appear as a potential miRNA target in the literature and mention that in the text.

We appreciate for the suggestion and reanalyzed candidate miRNAs using psRNATarget program (Xinbin Dai and Patrick X. Zhao, *Nucleic Acids Research*, 2011). We also analyzed small RNAs at PpCSP1 locus using “Locus Reporter”, which combines experimental data of small RNA-seq and degradome in *Physcomitrella* (Ceyda Corch et al., *Plant Cell*, 2015). In either case, we did not find any candidate miRNAs targeting *PpCSP1*.

We changed ‘We subsequently searched candidate miRNAs using the 3’ UTR as a BLAST query in the miRBase website (www.mirbase.org)’ to “We subsequently searched candidate miRNAs using the 3’ UTR as a query in the psRNATarget website (<http://plantgrn.noble.org/psRNATarget/>) and analyzed small RNAs at *PpCSP1* locus in Plant Small RNA Genes WebServer (https://plantsmallrnagenes.psu.edu/cgi-bin/Ppatens_Locus_Reporter)” in line 252-256 in this file.

Reviewer #4

The main questions raised in my previous review were: 1) what is the nature of the "reprogramming" seen in moss, and 2) can this be clearly separated from a role of PpCSP1 in cell cycle progression. In the revised version, I do not think these questions have been addressed satisfactorily.

In relation to question 1:

Looking at Supplementary video 1, I am not convinced that the apparent decrease in the expression of PpCSP1 in some apical cells is due simply to the cells moving out of the focal plane. Based on the bright field images, some apical cells are clearly out of focus but show strong PpCSP1 signal, whereas others are closer to the focal plane but show little expression (e.g. the apical cell just about to cross the lower edge of the image at the end of the video). In addition, the video was cropped in a way that did not allow me to check whether the pattern shown for normal protonemata (Supplementary video 2) is quickly established during regeneration from cut leaves.

Although the authors state that they could not detect oscillation of PpCSP1 expression during the cell cycle, the videos showing PpCSP1 expression during regeneration from leaves (Supplementary videos 1, 3, 4) do show a striking pattern with expression peaking just before or during cytokinesis, after which some of the protein apparently remained associated with the newly made cell wall. This pattern looks different from that seen in protonema cells (Supplementary video 2) and suggests a role associated with cell cycle progression.

We appreciate for the careful observation of this reviewer. We reexamined the movie and found that PpCSP1-Citrine localizes at the phragmoplast and the intensity of PpCSP1-Citrine signal gradually decreased during the successive cell divisions of the newly formed chloronema apical stem cells. We added Supplementary Fig. 3 and Supplementary Video 2 as well as the following sentences at line 142-149 in this file.

“After the protrusion, PpCSP1-Citrine signals localized more conspicuously at the phragmoplast than other parts in cytosol. The signals were dispersed in cytosol after cytokinesis with remaining signals at the cell septum. The signals at the phragmoplast decreased during subsequent cell divisions of chloronema apical stem cells (Supplementary Fig. 3; Supplementary Video 2). These indicates that PpCSP1 protein predominantly accumulates in the leaf cells facing the cut, accumulates during the reprogramming, gradually decreases after the reprogramming, and is maintained in stem cells.”

We also found that PpCSP1-Citrine signals increased around the protruded part of protonema cells and localized at the phragmoplast, and described as follows in line 150-153 in this file.

“When side branch cells initiated, PpCSP1-Citrine signals increased during protrusion and localized at the phragmoplast. The signals at the phragmoplast decreased during subsequent cell divisions (Supplementary Video 3).”

In reply to the question whether PpCSP1 might play a role in cell cycle progression rather than specifically in stem cell identity, the authors state that "All stem cells divide and there are no quiescent stem cells in *Physcomitrella* protonemata. So it is not possible to do the requested experiment". The experiment requested does not need to be confined to protonemata. If the authors can show at any stage of the moss life cycle examples of dividing cells that do not express PpCSP1, this would be sufficient to break the correlation between PpCSP1 and cell cycle progression.

The authors also state that "Instead, we added that increase and decrease of PpCSP1 levels in nPpCSP1-Citrine-nosT and quadruple deletion mutant lines, respectively did not cause effects in cell cycle progression in protonema growth in line 301 and line 336". However, I could not find the new data showing that altered PpCSP1 levels did not affect cell cycle progression.

Based on this reviewer's suggestion, we observed a gametophore tip where both stem cells and proliferating non-stem cells exist and found that PpCSP1 is expressed in proliferating cells including stem cells and non-stem cells (Supplementary Fig. 1e). We could not find any proliferating cells without PpCSP1 expression. On the other hand, we did not see any changes in cell division patterns of proliferating cells in quadruple disruptants and overexpression lines as shown in Supplementary Figure 8. As we mentioned above, we found that PpCSP1 localizes at the phragmoplast at the reprogramming leaf cells and the signals diminishes in newly formed chloronema apical stem cells. However, at this stage, it is not possible to connect PpCSP1 with cell cycle regulation during the reprogramming. Therefore, we added the following sentences in discussion.

“PpCSP1-Citrine signals localized at the phragmoplast when the reprogrammed leaf cells divide (Supplementary Fig. 3; Supplementary Video 2). The signals were maintained in the reprogrammed chloronema apical stem cells and diminished in the successive cell divisions, although the diminished signals were maintained in chloronema apical stem cells (Supplementary Fig. 3; Supplementary Video 2). These results suggest that PpCSP1 is involved in the cell cycle regulation during or after the reprogramming, as Lin28 promotes cell-cycle regulators and coordinates proliferative growth^{59,60}. However, increase and decrease of PpCSP1 levels in nPpCSP1-Citrine-nosT and quadruple deletion mutant lines, respectively did not change cell growth in protonema cells (Supplementary Fig. 8). It is a future problem whether PpCSP1 functions in cell cycle regulation during the reprogramming.” in line 392-402 in this file.

In relation to question 2:

Reprogramming normally refers to changes in gene expression that underlie a change in cell identity. For this reason, it would be important to show that changes in the expression of other stem cell markers are among the earliest effects of PpCSP1, before any visible changes in growth or cell cycle progression. If the only way to detect the re-establishment of stem cells in moss is through morphological changes, then the authors should not use the word re-programming in a way that suggests a process similar to the induction of iPSC by Oct4, Sox2 and Nanog in mammals.

We agree that we should not use “reprogramming” in a way that suggests a process similar to the induction of iPSC by Oct4, Sox2 and Nanog in mammals. In the previous versions, we mentioned the difference of molecular mechanisms of reprogramming between PpCSP1 and Lin28, even though both genes enhance reprogramming. To further stress on this point, we changed the last sentence of the abstract from “Taken together, these data demonstrate the positive role of PpCSP1 in reprogramming, which is similar to mammalian Lin28.” to “Taken together, these data demonstrate the positive role of PpCSP1 in reprogramming, which is similar to mammalian Lin28 but the molecular mechanisms are different.” in line 43-45 in this file.

Reviewers' comments:

Reviewer #4 (Remarks to the Author):

My main criticism of the paper was that it remained unclear whether PpCSP1 had a specific role in reprogramming or whether it functions in cell division. The authors now provide additional data that reinforces the possibility that PpCSP1 may have a more general role in cell division.

Although PpCSP1 expression during gametophore divisions is now shown in Figure S1e, the point of showing this image is not made in the text and remains unclear to the reader: the reason why PpCSP1 is associated with protonemata stem cells and plays a role in regeneration may be simply that this process requires cell division.

The authors mention a lack of direct evidence for effects in cell cycle progression as a reason to leave open the possibility of a function in reprogramming. However, I think their negative evidence is not strong: it is not sufficient to show the size of mature gametophores or a static image of protonemata, because these cannot separate the contribution of cell proliferation and cell growth to overall organ growth and do not reveal differences in the rate of cell division. It should be straightforward to use live imaging to compare, for example, cell cycle length in the protonemata of the wild type and after gain and loss of PpCSP1 function. Some of the required information may already be in their videos.

The claim that Lin28 could have a conserved role in stem cell functions would be more exciting if this role related to cell fate decisions; a general function in cell cycle progression would be in the same category as the role of CDKA activation during reprogramming (reference 9 in the paper). This caveat of the paper does not match the key point made in the abstract: "Here we show that the moss *Physcomitrella patens* Cold-Shock Domain Protein 1 (PpCSP1) regulates reprogramming of differentiated leaf cells to chloronema apical stem cells and shares conserved domains with the induced pluripotent stem cell factor Lin28 in mammals."

The association between Lin28 and the phragmoplast is very interesting, but as the authors noted, it suggests that the molecular/cellular function of Lin28 could be different in moss and animals – at the very least because the phragmoplast is a plant-specific structure. This uniqueness also somewhat undermines the point about cross-kingdom conservation. Related to conservation across kingdoms, the abstract says: "Taken together, these data demonstrate the positive role of PpCSP1 in reprogramming, which is similar to mammalian Lin28 but the molecular mechanisms are different". This sentence appears ambiguous: does "similar" refer to the positive role in reprogramming, or to the sequence similarity between Lin28 and PpCSP1? If the molecular mechanisms are different, what is the significance of the molecular similarity between PpCSP1 and Lin28?

Overall, I think that there is certainly something interesting about the role of PpCSP1 in reprogramming in moss, but is it premature to suggest that the similarity to Lin28 reflects

conservation or convergence in the regulation of stem cell functions between animals and plants.

Reply to Reviewers' comments:

Reviewer #4 (Remarks to the Author):

My main criticism of the paper was that it remained unclear whether PpCSP1 had a specific role in reprogramming or whether it functions in cell division. The authors now provide additional data that reinforces the possibility that PpCSP1 may have a more general role in cell division.

I appreciate this reviewers' critical comments to understand the function of PpCSP1. At first, I just want to confirm that "a specific role" means "an exact role" but does not mean "an exclusive role" in reprogramming, since Lin28 is not exclusively involved in reprogramming but functions in other processes as well as reprogramming (e.g. Viswanathan and Daley 2010 Cell 140: 445; Cho et al. 2012 Cell 151: 765; Madison et al. 2013 Genes Dev. 27: 2233; Zhang et al. Cell Stem Cell 19: 3).

In this version, we included Dr. Masaki Ishikawa, an expert on cell cycle regulation as a coauthor and performed experiments to show that PpCSP1 is exactly involved in reprogramming rather than a broad role in cell cycle progression. Previously, Ishikawa et al. (2011: Plant Cell 23: 2924) found that cell cycle progression is not required for reprogramming in *Physcomitrella* as the cell cycle inhibitor aphidicolin treatment arrests cell cycle but not reprogramming. So we added aphidicolin to the cut leaves of the *PpCSP1* transcript-increased line and the quadruple deletion mutant. Cell cycle was arrested with aphidicolin but the protrusion, as an indicator of the reprogramming, was enhanced in the *PpCSP1* transcript-increased line and was diminished in the quadruple deletion mutant. In addition, we succeeded to indicate that the duration of cell cycle in protonemata was not significantly changed among the wild type, deletion mutant, and transcript-increased line. These results indicate that PpCSP1 does not have a broad role in cell cycle progression, but functions in reprogramming.

Although PpCSP1 expression during gametophore divisions is now shown in Figure S1e, the point of showing this image is not made in the text and remains unclear to the reader:

the reason why PpCSP1 is associated with protonemata stem cells and plays a role in regeneration may be simply that this process requires cell division.

In the previous version, we added the following sentences in discussion (line 409 in the following tracking-change manuscript).

“In addition, PpCSP1 was expressed in both stem cells and proliferating non-stem cells in gametophore apices (Supplementary Fig. 1e). These results suggest that PpCSP1 is involved in the cell cycle regulation during or after the reprogramming, as Lin28 promotes cell-cycle regulators and coordinates proliferative growth^{59,60}.”

In addition, in this version, we added the following sentence in results (line 155 in the following tracking-change manuscript).

“This is reminiscent of Lin28 that regulates cell cycles in stem cells^{59,60}.”

The authors mention a lack of direct evidence for effects in cell cycle progression as a reason to leave open the possibility of a function in reprogramming. However, I think their negative evidence is not strong: it is not sufficient to show the size of mature gametophores or a static image of protonemata, because these cannot separate the contribution of cell proliferation and cell growth to overall organ growth and do not reveal differences in the rate of cell division. It should be straightforward to use live imaging to compare, for example, cell cycle length in the protonemata of the wild type and after gain and loss of PpCSP1 function. Some of the required information may already be in their videos.

We appreciate for the suggestion to examine the involvement of PpCSP1 in general cell cycle regulation. In this version, we performed quantitative analyses of the duration of cell cycles in the wild type, quadruple deletion mutant, and nPpCSP1-Citrine-nosT line with live-imaging videos. We could not find any differences of the duration of cell cycle among these lines. This indicates that PpCSP1 is not involved in general cell cycle progression, even though its expression is associated with cell cycle. We added Supplementary Fig. 8g and Supplementary Video 6 as well as the following sentences at lines from 310 in the following tracking-change manuscript.

“PpCSP1 was expressed in not only stem cells but also proliferating non-stem cells in gametophore apices (Supplementary Fig. 1e) and appeared to localize at the phragmoplast (Supplementary Fig. 3; Supplementary Video 2). These data suggest the possibility that PpCSP1 is not involved in the reprogramming but in general cell cycle progression. To examine this possibility, we analyzed the phenotype of the quadruple deletion mutant and the *PpCSP1* transcript-increased line in protonemata and gametophores. We could not distinguish the protonemata and gametophores of the quadruple deletion mutant and the transcript-increased line from those of wild type (Supplementary Fig. 8a-f). Moreover, the duration of cell cycles of protonemata of these lines was measured with time-lapse observation and we could not find any differences (Supplementary Fig. 8g; Supplementary Video 6). These results suggest that PpCSP1 does not play a major role in cell cycle progression in protonemata.

When we added a DNA synthesis inhibitor, aphidicolin to cut leaves, cell cycle reentry was arrested but leaf edge cells are protruded, indicating that cell cycle progression is not required for the reprogramming⁹ (Supplementary Fig. 9). To examine whether PpCSP1 regulates reprogramming regardless of cell cycle, we treated aphidicolin to the quadruple deletion mutant, *PpCSP1* transcript-increased line, and wild type, and compared their reprogramming phenotype. In the presence of aphidicolin, *ppcsp* quadruple deletion mutant and *PpCSP1* transcript-increased line exhibited attenuated and enhanced reprogramming, respectively as in the absence of the cell-cycle inhibitor (Supplementary Fig. 9). These indicate that PpCSP1 functions in the reprogramming independently of cell cycle progression.”

The claim that Lin28 could have a conserved role in stem cell functions would be more exciting if this role related to cell fate decisions; a general function in cell cycle progression would be in the same category as the role of CDKA activation during reprogramming (reference 9 in the paper). This caveat of the paper does not match the key point made in the abstract: “Here we show that the moss *Physcomitrella patens* Cold-Shock Domain Protein 1 (PpCSP1) regulates reprogramming of differentiated leaf cells to chloronema apical stem cells and shares conserved domains with the induced pluripotent stem cell factor Lin28 in mammals.”

We indicated that PpCSP1 functions in the reprogramming by the aphidicoline experiments

and kept this sentence in this version.

The association between Lin28 and the phragmoplast is very interesting, but as the authors noted, it suggests that the molecular/cellular function of Lin28 could be different in moss and animals – at the very least because the phragmoplast is a plant-specific structure. This uniqueness also somewhat undermines the point about cross-kingdom conservation.

Related to conservation across kingdoms, the abstract says: “Taken together, these data demonstrate the positive role of PpCSP1 in reprogramming, which is similar to mammalian Lin28 but the molecular mechanisms are different”. This sentence appears ambiguous: does “similar” refer to the positive role in reprogramming, or to the sequence similarity between Lin28 and PpCSP1? If the molecular mechanisms are different, what is the significance of the molecular similarity between PpCSP1 and Lin28?

Overall, I think that there is certainly something interesting about the role of PpCSP1 in reprogramming in moss, but is it premature to suggest that the similarity to Lin28 reflects conservation or convergence in the regulation of stem cell functions between animals and plants.

In the previous versions, we evaluated that PpCSP1 may have different functions in the reprogramming from Lin28 because of the phragmoplast localization as well as the different 3'UTR regulation, even though both PpCSP1 and Lin28 positively regulate the reprogramming. However, in this version, we succeeded to show that PpCSP1 functions in the reprogramming before cytokinesis and would like to propose the necessity of further investigation of both RNA-binding proteins to explore their general role in cellular reprogramming. Also, the possible involvement of PpCSP in cell cycle reentry during the reprogramming may be related to Lin28 functions to promote cell-cycle regulators and coordinates proliferative growth as we mentioned in the last two paragraphs in discussion.

Therefore, we changed the last sentence of the abstract as follows.

From “Taken together, these data demonstrate the positive role of PpCSP1 in reprogramming, which is similar to mammalian Lin28 but the molecular mechanisms are different”.

To “Taken together, these data demonstrate the positive role of PpCSP1 in reprogramming, which is similar to the function of mammalian Lin28”.

REVIEWERS' COMMENTS:

Reviewer #4 (Remarks to the Author):

The authors added new experiments showing that loss and gain of function of PpCSP1 do not affect general cell cycle progression and that PpCSP1 facilitates the conversion of leaf cells to chloronema even when cell cycle progression is blocked. These results address my previous concerns and strengthen the link between PpCSP1 and the control of cell fate.

As a last suggestion, the text could be tidied up a bit. For example, the Discussion is long and in some places repetitive (compare lines 371-372 with 428-429). Minor grammar corrections are also required.

Reply to Reviewers' comments:

Reviewer #4 (Remarks to the Author):

As a last suggestion, the text could be tidied up a bit. For example, the Discussion is long and in some places repetitive (compare lines 371-372 with 428-429). Minor grammar corrections are also required.

We appreciate for the reviewer's comment and removed the repetitive sentences from the Discussion. The manuscript was English-edited by Prof. Philip Benfey, a native American.